# PABPN1 gene therapy for oculopharyngeal muscular dystrophy

A. Malerba[1,*], P. Klein[2,*], H. Bachtarzi[1,†], S.A. Jarmin[1], G. Cordova[2], A. Ferry[2,3], V. Strings[4],
M. Polay Espinoza[2], K. Mamchaoui[2], S.C. Blumen[5], J. Lacau St Guily[2,6], V. Mouly[2], M. Graham[4],
G. Butler-Browne[2], D.A. Suhy[4], C. Trollet[2,**] & G. Dickson[1,**]

Oculopharyngeal muscular dystrophy (OPMD) is an autosomal dominant, late-onset muscle disorder characterized by ptosis, swallowing difficulties, proximal limb weakness and nuclear aggregates in skeletal muscles. OPMD is caused by a trinucleotide repeat expansion in the *PABPN1* gene that results in an N-terminal expanded polyalanine tract in polyA-binding protein nuclear 1 (PABPN1). Here we show that the treatment of a mouse model of OPMD with an adeno-associated virus-based gene therapy combining complete knockdown of endogenous PABPN1 and its replacement by a wild-type PABPN1 substantially reduces the amount of insoluble aggregates, decreases muscle fibrosis, reverts muscle strength to the level of healthy muscles and normalizes the muscle transcriptome. The efficacy of the combined treatment is further confirmed in cells derived from OPMD patients. These results pave the way towards a gene replacement approach for OPMD treatment.

[1] School of Biological Sciences, Royal Holloway, University of London, Egham Hill, Egham, TW20 0EX Surrey, UK. [2] Sorbonne Universités, UPMC Univ Paris 06, UM76, INSERM U974, Institut de Myologie, CNRS FRE3617, 47 bd de l'Hôpital, 75013 Paris, France. [3] Sorbonne Paris Cité, Université Paris Descartes, 75006 Paris, France. [4] Benitec Biopharma, 3940 Trust Way, Hayward, California 94545, USA. [5] Department of Neurology, Hillel Yaffe Medical Center, Hadera and Rappaport Faculty of Medicine, The Technion, 1 Efron Street, Haifa 31096, Israel. [6] Department of Otolaryngology-Head and Neck Surgery, Faculty of Medicine and University Pierre-et-Marie-Curie, Paris VI, Tenon Hospital, Assistance Publique des Hopitaux de Paris, 75252 Paris, France. * These authors contributed equally to this work. ** These authors jointly supervised this work. † Present address: ERA Consulting (UK) Ltd. (European Regulatory Affairs), Twelvetrees Crescent, London E3 3JG, UK. Correspondence and requests for materials should be addressed to C.T. (email: capucine.trollet@upmc.fr) or to G.D. (email: g.dickson@rhul.ac.uk).

Oculopharyngeal muscular dystrophy (OPMD) is an autosomal dominant, degenerative muscle disease affecting about 1/100,000 people in Europe that usually presents in the fifth decade of life[1]. OPMD is mainly characterized by progressive eyelid drooping, swallowing difficulties (as the pharyngeal muscles are mostly affected) and proximal limb weakness[2].

Individuals affected by OPMD have a mutation in the *PABPN1* gene, coding for polyA-binding protein nuclear 1 (PABPN1)[3], an ubiquitously expressed polyadenylation factor involved in many biological processes[4]. PABPN1 stimulates the polydenylate polymerase and controls the poly(A) tail length on RNA transcripts[5,6]. PABPN1 regulates the use of alternative polyadenylation sites[7,8], which in turn affects mRNA levels and stability. PABPN1 is also involved in the long non-coding RNA[9] and small nucleolar RNA processing[10], and in the nuclear surveillance that leads to hyperadenylation and decay of RNA[11]. Finally, two recent studies highlight the role for PABPN1 in splicing regulation[12,13]. In OPMD, mutated PABPN1 has an abnormal expansion of alanine-encoding (GCN)n trinucleotide repeat in the coding region of exon 1 with 11–18 repeats instead of the normal 10 present in unaffected individuals[3,14]. The resulting protein (expPABPN1) is misfolded and prone to aggregation in nuclear insoluble aggregates and this is considered to be the main histopathological hallmark of the disease[15,16]. However, like in many neurodegenerative diseases[17,18], it is still unclear whether these nuclear aggregates have a pathological function or a protective role in OPMD as a consequence of a cellular defence mechanism[19–21].

No cure is currently available to arrest the disease. Surgical cricopharyngeal myotomy is the only treatment available to improve swallowing in these patients, but the pharyngeal musculature still undergoes progressive degradation leading to severe swallowing impairment, pulmonary infections and choking[2,22]. Molecules potentially ameliorating the OPMD phenotype (for example, trehalose and doxycycline) have been tested in a mouse model of OPMD, where reduction in muscle weakness was observed[23,24]. The use of trehalose is currently in phase II clinical trial (NCT02015481 on clinicaltrials.gov). Other pharmacological strategies currently under pre-clinical investigation include antiprion drugs like 6-aminophenanthridine and guanabenz, and targeting the expPABPN1 with intracellular antibodies[25–27]. However, none of these strategies directly correct the genetic defect of OPMD patients. A recent phase I/IIa clinical trial that employed transplantation of unaffected autologous myoblasts in combination with the cricopharyngeal myotomy ameliorated the pathology of OPMD patients with a cell dose-dependent improvement in swallowing[28]. However, the injected cells still carry the mutation and thus may be susceptible to the same OPMD phenotype.

The most commonly used murine model of OPMD, the A17 mouse model, expresses a bovine expPABPN1 with 17 alanine residues under the control of the human alpha actin muscle-specific promoter. In heterozygous mice, this model recapitulates most of the features of human OPMD patients, including progressive atrophy and muscle weakness associated with nuclear aggregates of insoluble PABPN1 (refs 21,24,29).

Adeno-associated virus (AAV) is currently the most promising viral vector for *in vivo* gene therapy applications due to its non-pathogenicity, the natural efficient infection in primates for some serotypes and the negligible risk of insertional mutagenesis[30,31]. Within the 4.7 kb vector packaging capacity, AAV vectors can accommodate both transgenes and shRNA/miRNA expression cassettes[32]. Recent clinical studies based on localized or systemic administration of AAV vectors for gene expression in humans have shown potential for the treatment of human monogenic diseases, such as haemophilia, retinal degeneration, metabolic disorders, muscular dystrophies and degenerative diseases[33–38]. The use of AAV-mediated RNAi therapeutics is receiving great attention in a wide range of pathologies[32], including dominant genetic disorders that involve gain-of-function mechanisms[39] and protein aggregate disorders for which a silencing approach has shown some promise in pre-clinical studies[40,41].

Here we demonstrate a dual gene therapy approach in a mouse model of OPMD by using RNA interference (RNAi) to inhibit the expression of all endogenous (that is, both wild type and expanded) PABPN1 and express a sequence-optimized normal PABPN1 resistant to RNAi-induced cleavage. The simple depletion of the endogenous (wild type and expanded) PABPN1 in mouse muscles effectively abolishes the nuclear aggregates, but induces muscle degeneration. Furthermore, AAV-mediated expression of a sequence-optimized human PABPN1 alone does not significantly improve the pathology either. On the contrary, the simultaneous application of the two treatments prevents muscle degeneration, markedly increases muscle strength, normalizing the specific muscle force to that of unaffected muscles, and essentially normalizes muscle gene expression profiles. The efficacy of the combined treatment was also verified in cells derived from OPMD patients, where the expression of transduced normal PABPN1 rescued the survival of cells where the endogenous PABPN1 was downregulated. Overall, our data indicate that PABPN1 aggregation induces, in OPMD, both a gain of toxic function and a loss of physiological function and strongly support the application of a dual gene therapy approach as a novel treatment for OPMD in humans.

## Results

**PABPN1 knockdown and replacement validation *in vitro*.** DNA-directed RNA interference (ddRNAi) uses the cell's own transcriptional machinery to produce short double-stranded hairpin RNA (shRNA), which when processed into siRNA can degrade specific target mRNA. In order to ensure efficient knockdown of the PABPN1, three shRNA sequences (designated sh-1, sh-2 and sh-3) were designed. Sequences sh-1 and sh-3 target *PABPN1* mRNA in regions of conserved identity between mouse, bovine and human species, while sh-2 specifically binds bovine and human PABPN1. In order to provide a strong knockdown of *PABPN1* transcript levels, we assembled the coding sequences for the three shRNAs into a tricistronic expression cassette (called shRNA3X), with each hairpin RNA driven by a different polymerase III promoter (U61, U69 and H1, respectively) (Fig. 1a). As for *in vivo* muscle gene therapy applications the use of AAV vectors is very efficient, the tricistronic shRNA3X cassette was cloned into an AAV backbone. We also generated two separate plasmids harbouring a single-stranded (ss) AAV expression cassette carrying either the human codon-optimized PABPN1 sequence (optPABPN1) tagged with a MYC epitope or the mutated expanded human PABPN1 complementary DNA (cDNA) (expPABPN1) coding sequence tagged with a FLAG epitope, both under the control of the muscle-specific SPc5-12 promoter. With codon optimization, the redundancy of the genetic code was exploited to largely modify the nucleic acid sequence of PABPN1 (that is, 230 out of 921 nucleotides mismatches, corresponding to 25% difference) and confer resistance of optPABPN1 to the expressed shRNA sequences. To verify that shRNA3X was able to knock down PABPN1 (both wild-type and mutated version of PABPN1) while maintaining unaffected levels of optPABPN1, we transfected HEK293T cells with the recombinant expression plasmid harbouring shRNA3X alone or used this plasmid in co-transfection

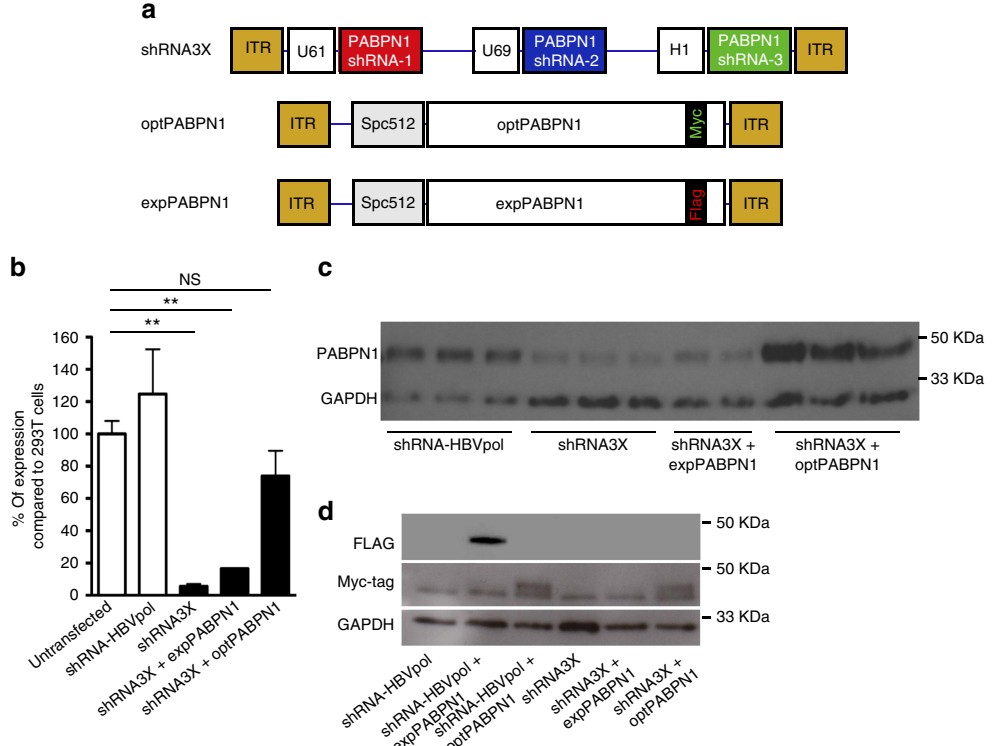

**Figure 1 | The tricistronic construct efficiently silences endogenous normal and expanded PABPN1 without affecting optPABPN1 expression *in vitro*.**
(**a**) Three constructs were cloned into pAAV vectors and used in this study: a tricistronic shRNA construct including hairpins sh-1, sh-2 and sh-3 driven each by a different polymerase III promoter (U61, U69 and H1, respectively), the sequence-optimized PABPN1 (optPABPN1) driven by the SPc5-12 promoter and tagged with a MYC-tag, and the human expanded PABPN1 (expPABPN1) driven by the SPc5-12 promoter and tagged with a FLAG-tag. (**b**) HEK293T cells were transfected with 4 µg per well of pAAV-shRNA3X in triplicate with or without AAV plasmids expressing expPABPN1 or optPABPN1. Untransfected cells and cells transfected with AAV plasmid expressing shRNA for HBVpol were used as a control. Seventy-two hours post transfection, samples were collected and PABPN1 was detected by western blot and quantified by densitometric analysis using ImageJ software. Transfection was performed twice. PABPN1 expression in each condition was normalized by GAPDH expression level and then by the value of untransfected cells. Transfection with a plasmid expressing shRNA3X induced efficient PABPN1 knockdown even when expPABPN1 was co-expressed compared to untransfected cells. The co-expression with optPABPN1 restored PABPN1 expression to the normal level. (**c**) Representative image of a western blot used for these analyses. (**d**) FLAG is not detected when samples are transfected with shRNA3X. However, in samples prepared from cells transfected with pAAV-opPABPN1, MYC-Tag is detected by western blot. Detection of MYC-Tag shows that optPABPN1 is resistant to degradation by shRNA3X *in vitro*. Data are presented as mean ± s.e.m., $n = 4$ (shRNA3X + optPABPN1) or $n = 6$ (all the other groups). One-way ANOVA test with Bonferroni *post-hoc* test *$P < 0.05$, **$P < 0.01$, NS, not significant.

experiments with the plasmids expressing expPABPN1 or optPABPN1. A negative control shRNA, designed to target HBV polymerase gene (HBVpol), was also transfected in parallel wells. Samples were collected 72 h post transfection and PABPN1 expression levels were detected by western blot analysis. The shRNA3X construct induced efficient knockdown of PABPN1 even when expPABPN1 was co-expressed, achieving about 95% knockdown in PABPN1 expression (Fig. 1b–d). Importantly, detection of MYC-tag by western blot in samples treated with plasmids expressing optPABPN1 demonstrated that shRNA3X induced efficient knockdown of the expPABPN1 protein, leaving the expression of optPABPN1 protein unaltered (Fig. 1d).

Overall, these data demonstrate that the tricistronic construct knocks down both wild-type and expanded PABPN1. Furthermore, when co-delivered with shRNA3X, the human sequence-optimized PABPN1 is resistant to degradation and results in expression of a normal PABPN1.

**PABPN1 knockdown and replacement in OPMD mouse muscle.**
The corresponding AAV vectors were produced in HEK293T cells by a double-transfection-based protocol and formulated for

*in vivo* injection. We selected serotype 8 and 9 AAV vectors based on previous studies that demonstrated high levels of transduction and expression in skeletal muscle[42,43]. To ensure robust PABPN1 knockdown, shRNA3X was delivered by self-complementary AAV8 (scAAV-shRNA3X) that have been demonstrated to achieve a more rapid and efficient transgene expression than their single-stranded counterparts[44]. The expression cassette for optPABPN1 was packaged as a single-stranded construct into AAV9 (ssAAV-optPABPN1) as this serotype has a better packaging, resulting in higher titres (unpublished observations). The A17 mouse model of OPMD overexpresses bovine expPABPN1 along with the healthy murine version and displays a progressive muscle atrophy and weakness associated with nuclear PABPN1 aggregates[21,24]. AAVs were administered by intramuscular injection in *Tibialis anterior* (TA) muscles of 10-12-week-old A17 mice at a concentration of either $2.5 \times 10^{10}$ viral particles (vp) of scAAV-shRNA3X or $1.3 \times 10^{11}$ ssAAV-optPABPN1 (respectively referred as shRNA3X and optPABPN1), or as a combination of the two AAVs at the same concentrations. Saline was injected in TA of age-matched A17 and FvB (wild-type control strain) mice. Eighteen weeks post injection, muscles were collected and RNA and proteins

extracted. Quantitative RT-PCR (qRT-PCR) showed a 35 and 45% decrease in endogenous PABPN1 expression in muscles treated with either shRNA3X or shRNA3X + optPABPN1, respectively, compared to saline-injected A17 muscles (Fig. 2a). Western blot analysis detected a more profound inhibition (that is, 80% knockdown) of expPABPN1 protein expression in muscles treated with the shRNA3X as compared to saline-injected A17 mice, and even greater levels of inhibition of PABPN1

protein levels (that is, approximately 90%) were noted upon co-delivery of shRNA3X and optPABPN1 (Fig. 2b,c and Supplementary Fig. 1a). Both qRT-PCR and western blot further indicated that optPABPN1-MYC was successfully expressed in muscles treated with optPABPN1 or the combination of shRNA3X and optPABPN1 (Fig. 2d,e and Supplementary Fig. 1b). We did not detect a difference in MYC-tag quantification between muscles treated with

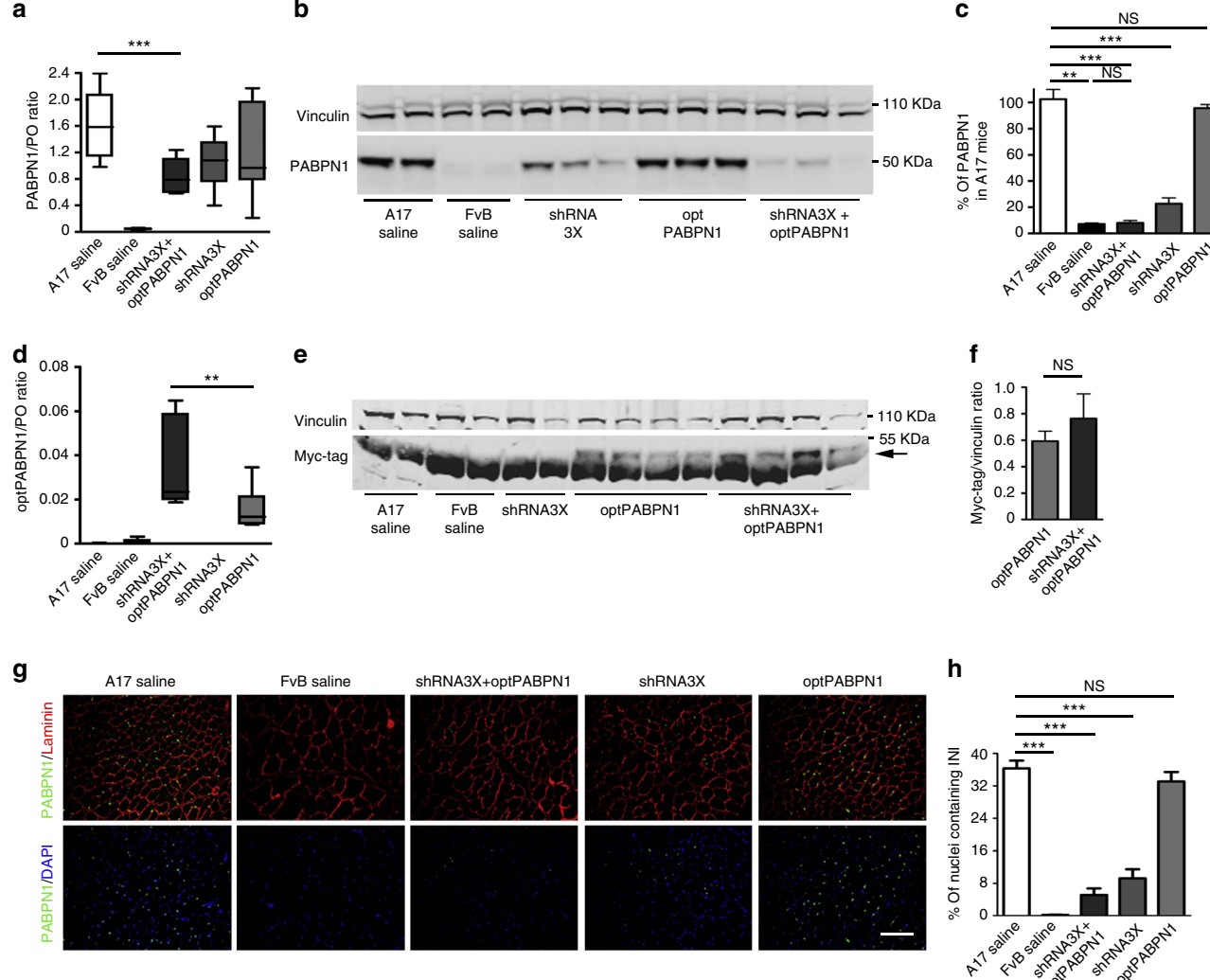

**Figure 2 | Tricistronic shRNA delivered *in vivo* by AAV vectors efficiently downregulates endogenous PABPN1 without affecting optPABPN1 expression *in vivo*.** Eighteen weeks after intramuscular AAV injections, TA muscles were collected and expression of PABPN1 was analysed. (**a**) qRT-PCR analysis of *PABPN1* mRNA normalized to the housekeeping gene *RplP0* mRNA shows a statistically significant decrease in PABPN1 expression in muscles injected with shRNA3X and optPABPN1 compared with muscles injected with saline. (**b**) Western blot for PABPN1 expression shows that the treatments with AAV-shRNA3X alone or in combination with AAV-optPABPN1 significantly knock down the endogenous PABPN1. (**c**) Densitometric analysis of western blot for PABPN1 detection shows statistically significant reduction in protein expression when muscles are treated with shRNA3X (with or without optPABPN1 expression) compared to the level of A17 muscles. PABPN1 expression was normalized to the relative vinculin expression. (**d**) qRT-PCR analysis for *optPABPN1* mRNA normalized to the housekeeping gene *Rplp0* mRNA shows significant expression in muscles where AAV-optPABPN1 was injected (with or without shRNA3X co-expression). (**e**) Representative western blot for MYC-tag shows that the epitope is detected in all muscles treated with AAV-optPABPN1 alone or in combination with AAV-shRNA3X. The arrow shows the band detected at the correct molecular weight. (**f**) Averages obtained by densitometric analysis of MYC-tag detected in samples treated with AAV-optPABN1 only or with both AAV-shRNA3X and optPABPN1 viral vectors indicates that shRNA3X does not affect optPABPN1 protein amount when co-expressed in muscles injected with both vectors. Two-tailed Student *t*-test, $n = 8$, NS, not significant. (**g**) Detection of PABPN1 inclusions (green) and Laminin (red) by immunofluorescence in sections of treated muscles. Sections were pre-treated with 1 M KCl to discard all soluble PABPN1 from the tissue. Nuclei are counterstained with DAPI (blue). Bar, 200 µm. (**h**) Quantification of percentage of nuclei containing INIs in muscle sections indicates that treatments with either AAV-shRNA3X or both AAV-shRNA3X and optPABPN1 significantly reduce the amount of INIs to about 10% and 5%, respectively, compared to saline-injected A17. For **a,c,d,h**, $n = 6$ (saline-treated A17 or FvB muscles) or $n = 8$ (all the other groups). Data are presented as mean ± s.e.m. One-way ANOVA test with Bonferroni post-doc test **$P < 0.01$, ***$P < 0.001$, NS, not significant.

optPABPN1 and muscles treated with shRNA3X + optPABPN1, confirming that *in vivo* optPABPN1 is resistant to the PABPN1 degradation provided by shRNA3X (Fig. 2e,f).

Quantitative PCR (qPCR) also confirmed the expression of shRNAs from the tricistronic construct in shRNA3X- and shRNA3X + optPABPN1-treated muscles. Interestingly, a statistically significant difference in shRNA levels was noted between muscles treated with shRNA3X only versus muscles treated with a combination of both vectors (with a lower amount in the shRNA3X-treated group) (Supplementary Fig. 1c). The effects of PABPN1 knockdown were also assessed by analysing the amount of insoluble aggregates in the myonuclei, the main pathological hallmark of the disease, which are formed due to expPABPN1 expression. Insoluble nuclear aggregates were readily detected in about 35% of myonuclei in A17 mice, while FvB myonuclei contained virtually no aggregates. While the expression of optPABPN1 did not modify the amount of insoluble aggregates in A17 muscles, the treatment with either shRNA3X alone or the combination of shRNA3X and optPABPN1 greatly decreased the amount of myonuclei containing aggregates to 10 and 5% of myonuclei, respectively (Fig. 2g,h).

These data show that shRNA3X delivered by AAV efficiently downregulates PABPN1 *in vivo* and significantly decreases the amount of insoluble PABPN1 aggregates. Importantly, a codon-optimized PABPN1, resistant to shRNA3X-induced degradation, can be co-delivered and expressed by AAV.

**PABPN1 knockdown alone has a detrimental effect in muscle**. Analysis of the shRNA3X-treated muscles group demonstrated that depletion of endogenous PABPN1 by shRNAs induced significant muscle degeneration/regeneration, as indicated by the increased presence of centrally nucleated fibres (50% in shRNA3X-treated muscles compared to 5% in saline-injected A17 mice) (Fig. 3a,c) as well as the increased levels of embryonic and neonatal myosins (eMHC and neoMHC, respectively), which are expressed when muscle fibres regenerate (Fig. 3b,d). Notably, the co-expression of optPABPN1 with shRNA3X in the A17 mice was able to prevent the muscle fibre degeneration and to maintain the level of centrally nucleated fibres as well as the low level of MHC expressed in regenerating fibres (Fig. 3a–d). In order to analyse the effect of *in vivo* PABPN1 depletion in wild-type mice, where only the normal endogenous level of PABPN1 is expressed, TA muscles of C57BL/6 mouse were injected with $2.5 \times 10^{10}$ vp of scAAV-shRNA3X or a control AVV expressing green fluorescent protein (ssAAV-GFP). Three months post injection, massive muscle degeneration was observed, as evidenced by the presence of more than 60% centrally nucleated fibres, increased embryonic myosin expression, decreased muscle weight and impaired muscle strength in shRNA3X-treated muscles as compared to GFP-treated TA muscles (Fig. 3e–j).

These data indicate that PABPN1 depletion in skeletal muscle is detrimental and induces extensive muscle degeneration. The co-expression of codon-optimized human PABPN1 prevents the shRNA-induced muscle degeneration and normalizes the muscle phenotype to that of untreated A17 mice.

**Histological and functional improvement in OPMD mouse muscle**. In muscles co-treated with shRNA3X and optPABPN1, we observed a significant reduction in fibrotic tissue compared with the saline-injected A17 muscles as monitored by staining for some of the main components of the extracellular matrix: collagens I, III and VI and fibronectin were detected by immunofluorescence and Sirius red staining (Fig. 4a,b and Supplementary Fig. 2). Myofibre cross-sectional area (CSA) analysis also indicated that while injection of shRNA3X alone did not modify the size distribution of myofibres, the treatment with optPABPN1 alone or in combination with shRNA3X markedly increased the myofibre CSA (Fig. 4c and Supplementary Fig. 3). Before collecting the muscle tissues, TA muscle strength of treated mice were analysed by *in situ* force measurement. No difference was detected in resistance to eccentric contraction between A17 and wild-type FvB mice, suggesting that the expPABPN1 expression does not affect proteins associated to the myofibre sarcolemma (Supplementary Fig. 4). However saline-injected muscles of A17 mice have a reduced muscle weight and absolute maximal tetanic force compared with muscles of healthy mice (Fig. 4d,e). Only the dual treatment with shRNA3X and optPABPN1 significantly increased the absolute maximal tetanic force generated by the muscles while treatment with either shRNA3X or optPABPN1 alone did not affect the muscle force. Surprisingly, no substantial increase of muscle weight was observed for any of the treatments (Fig. 4d) compared to saline-injected A17 muscles. The normalization of the maximal force by the muscle weight provides a measure of the muscle strength per unit of tissue called specific maximal force (Fig. 4f). The specific maximal force of muscles co-treated with the two vectors shRNA3X and optPABPN1 was restored to the value of healthy muscles, demonstrating that myofibres of treated muscles were as strong as those of wild-type mice. The concomitant reduction in muscle fibrosis and the small, but significant, increase in myofibre CSA may explain the unchanged muscle weight and at least partially account for the muscle strength improvement we observed in treated muscles (Fig. 4f).

Overall these results show that while single treatments inducing either the downregulation of PABPN1 or the expression of normal PABPN1 protein do not ameliorate the pathology, the combined treatment substantially improves the overall physiology and increases muscle strength restoring the specific maximal force to the wild-type level.

**Gene expression profile correction in OPMD mouse muscle**. Endogenous expPABPN1 expression in A17 mice substantially modifies the muscle transcriptome compared to wild-type mice[21,29]. The effect of the gene therapy strategy on the A17 muscle transcriptome was investigated using microarray analysis to identify the number of genes dysregulated ($>1.5$ fold change and $P<0.05$) in the A17 model (Fig. 5a). As compared to saline-treated FVB mice, 452 genes were upregulated (Fig. 5b), while an additional 413 of the sampled genes were downregulated (Fig. 5c). Treatment with either the shRNA3X or the optPABPN1 resulted in only modest normalization of the dysregulated genes, while the combined treatment with shRNA3X and optPABPN1 vectors completely normalized the dysregulated genes. In other words, of the original 865 genes that were dysregulated by more than 1.5 fold, only 12 genes remained significantly different following the combined shRNA3X and optPABPN1 treatment in the muscles of A17 mice (Fig. 5a–c). This means that greater than 98% of deregulated transcripts were normalized by the dual treatment in the muscles.

Overall these results show that the combined treatment leads to a normalization of the muscle transcriptome.

***Ex vivo* correction of OPMD patient myoblasts**. In order to verify the beneficial effect of the PABPN1 gene therapy strategy in the context of the human target, we used a similar approach in muscle cells derived from OPMD patients where expPABPN1 is expressed. Because AAV is highly inefficient at transducing primary myoblasts *in vitro*, the expression cassettes from the shRNA3X and optPABPN1 constructs were subcloned in lentivirus (LV) backbones that co-expressed the GFP reporter gene to detect transduced cells. Constructs used included the tricistronic

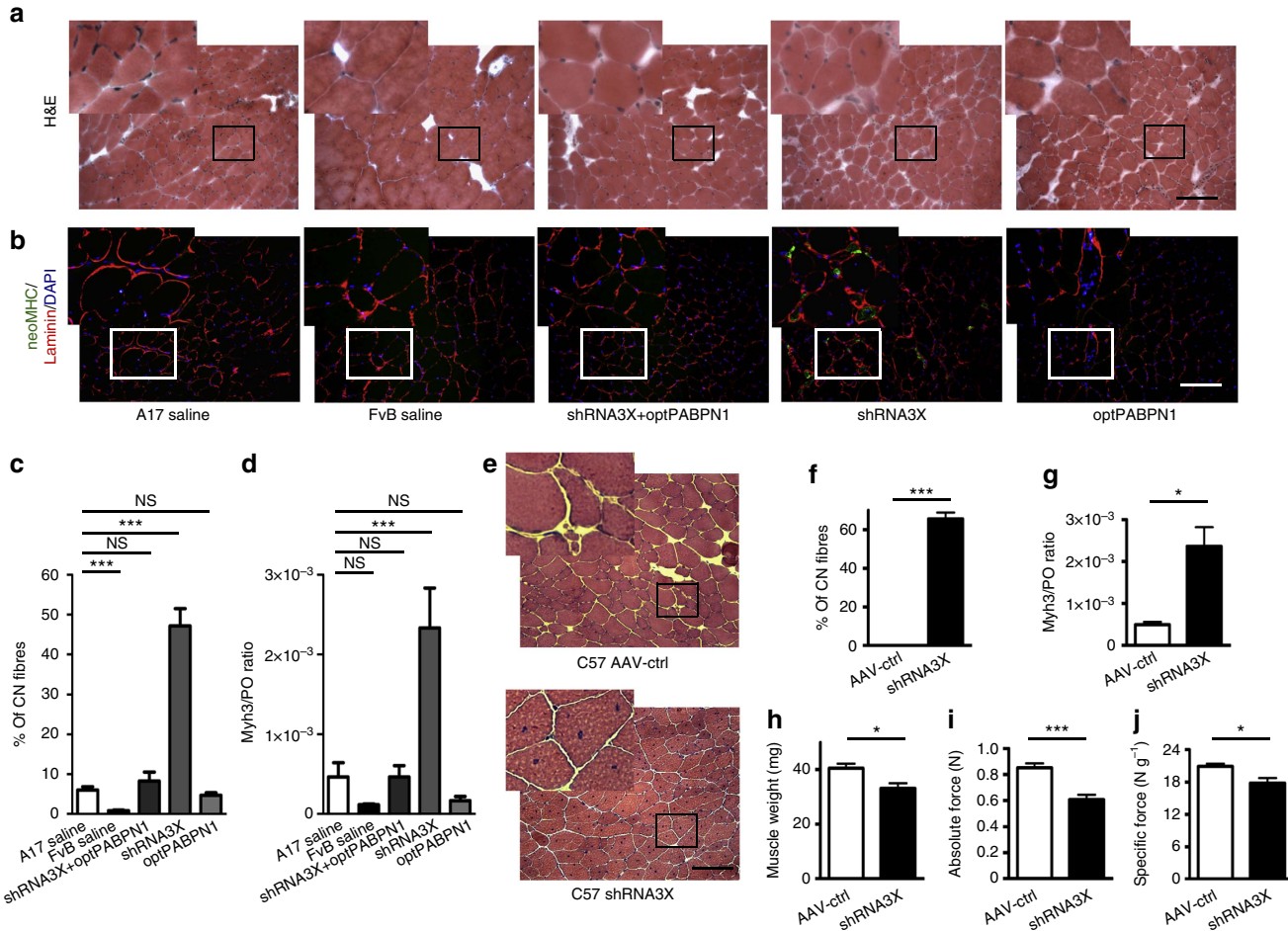

**Figure 3 | *In vivo* PABPN1 depletion induces muscle degeneration that can be prevented by co-optPABPN1 expression. (a)** H&E staining of sections from treated muscles of 30-week-old mice indicates that depleting endogenous PABPN1 in A17 muscles increases the amount of centrally nucleated fibres. Bar, 200 μm. **(b)** Immunostaining for neonatal MHC (green), laminin (red) and DAPI (blue) shows small regenerating neonatal MHC-positive myofibres in muscles treated with shRNA3X only. Bar, 200 μm. **(c)** Quantification of centrally nucleated fibres indicates that co-injecting AAV-optPABPN1 preserves the amount of fibres with central nuclei to the same level observed in saline-injected A17 muscles, thus preventing muscle degeneration. **(d)** qRT-PCR analysis of *Myh3* mRNA encoding embryonic MHC normalized to the housekeeping gene *RplP0* mRNA confirms the regeneration process in shRNA3X-treated muscles. **(e)** H&E staining of sections from treated muscles shows that depleting endogenous PABPN1 in WT muscles increases the amount of centrally nucleated fibres. Bar, 200 μm. **(f)** Quantification of centrally nucleated fibres in treated muscles indicates that PABPN1 depletion in WT muscles induce muscle degeneration, as for A17 treated muscles. **(g)** qRT-PCR analysis of *Myh3* mRNA encoding embryonic MHC normalized to the housekeeping gene *RplP0* mRNA shows that a regeneration process is ongoing in shRNA3X treated muscles. **(h)** PABPN1 depletion in WT muscles reduces muscle weight compared to contralateral muscles. **(i,j)** PABPN1 inhibition decreases both absolute maximal tetanic and specific maximal force generated by TA muscles of wild-type mice. Data are presented as mean ± s.e.m. For **c,d**, one-way ANOVA test with Bonferroni *post-hoc* test, $n = 6$ (saline-treated A17 or FvB muscles) or $n = 8$ (all the other groups). For **f-j**, two-tailed Student *t*-test, $n = 6$, *$P < 0.05$, ***$P < 0.001$, NS, not significant.

shRNA3X cassette, a construct expressing the control HBVpol-shRNA, a construct expressing both the tricistronic shRNA3X and optPABPN1, and finally a construct expressing the control HBVpol-shRNA and optPABPN1 (Supplementary Fig. 5). A strong knockdown of expPABPN1 was observed in HEK293T cells after co-transfection of the shRNA3X LV construct with expPABPN1 expression vector (Supplementary Fig. 6a). However, PABPN1 knockdown in HEK293T cells was associated with a strong reduction in cell viability after 2 weeks in culture (Supplementary Fig. 6b,c). OPMD myoblasts were then transduced with the corresponding LVs. GFP epifluorescence was used to monitor transduced cells. Similar to the transfection in HEK293T cells, PABPN1 inhibition by shRNA3X was associated to a substantial cell death of transduced cells: cell survival of OPMD myoblasts treated with shRNA3X was severely affected 4 days after transduction and reduced to undetectable levels 10 days after treatment (Fig. 6a). On the contrary, cells transduced

with the dual cassette (shRNA3X and optPABPN1) were healthy and proliferating 10 days after transduction (Fig. 6b), showing that replacement of endogenous PABPN1 levels with optPABPN1 expression was sufficient to rescue cell survival of the human OPMD myoblasts.

These data confirm that inhibition of endogenous PABPN1 expression is detrimental in a human cellular context and, most importantly, in human myoblasts isolated from OPMD patients. Notably, human optPABPN1 expression rescues cell survival from the toxic effect of PABPN1 inhibition.

## Discussion

No therapeutic intervention is currently available for patients with OPMD, a dominant genetic disease characterized by the formation of cellular inclusions as observed in other so called 'aggregopathies' (for example, Parkinson's, Alzheimers's,

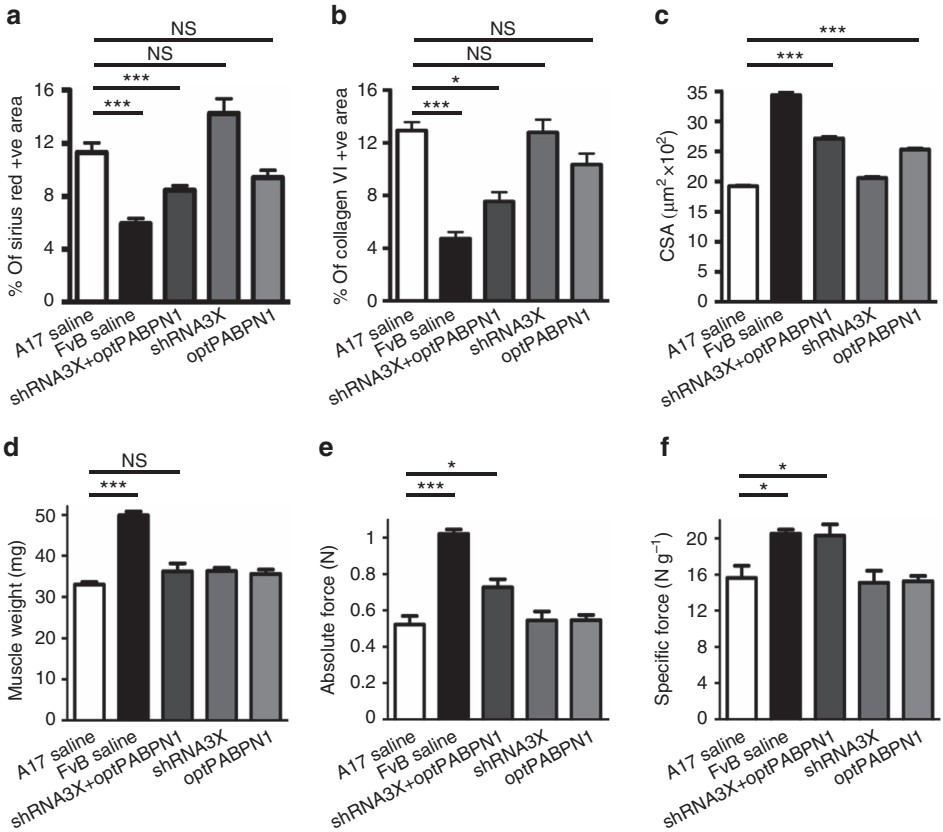

**Figure 4 | *In vivo* co-administration of AAV-shRNA3X and AAV-optPABPN1 diminishes muscle fibrosis and myofibre atrophy and improves the functionality of treated muscles.** (**a**,**b**) Morphometric evaluation of Sirius red staining (**a**) or collagen VI immunostaining (**b**) in treated muscles of 30-week-old mice shows a significant reduction in fibrosis in muscles treated with AAVs expressing a combination of shRNA3X and optPABPN1 compared to saline-treated muscles. (**c**) The average of myofibre sizes per group shows that myofibres of muscles treated with AAV-optPABPN1 only or in combination with AAV-shRNA3X are larger than myofibres of muscles treated with saline. (**d**) Eighteen weeks after AAVs injection no difference was detected in the muscle weight of AAV-treated muscles compared to saline-injected TA of A17 mice. (**e**) Maximal force generated by TA muscles of treated mice was measured by *in situ* muscle physiology, co-injecting AAV-shRNA3X and AAV-optPABPN1 significantly increased the absolute maximal tetanic force generated by TA muscles. (**f**) Normalization of maximal force by muscle weight provides a measure of the muscle strength per unit of skeletal muscle called specific maximal force. The co-injection of the two AAVs normalized the specific maximal force of TA muscles to the level detected by wild-type muscles. Data are represented as mean ± s.e.m. **a**–**c**: $n = 6$ (saline-treated A17 or FvB muscles) or $n = 8$ (all the other groups). **d**–**f**: $n = 8$ (saline-treated A17 or FvB muscles) or $n = 10$ (all the other groups). One-way ANOVA test with Bonferroni *post-hoc* test, $*P < 0.05$, $***P < 0.001$, NS, not significant.

Huntington's diseases). Currently, OPMD patients are surgically treated for the clinical manifestations of the disease, mainly by blepharoplasty to correct the ptosis, and cricopharyngeal myotomy to improve dysphagia. Like other protein aggregation disorders, pre-clinical and clinical studies based on the inhibition of aggregation formation or the clearance of existing aggregates are also being developed for OPMD[23,25–27,45]. However, none of these treatments directly corrects the genetic defect responsible for the diseased phenotype in the OPMD patients. Gene therapy holds the promise of providing a permanent cure for such diseases that are untreatable or treatable but not curable with conventional medicines[46]. As previously suggested by others[47], a strategy that could specifically impact the mutant allele would be suitable for OPMD, but the small, abnormal 100% GC-rich expansion in the mutant PABPN1 gene cannot easily be targeted without impacting the wild-type PABPN1 function. As a consequence, autosomal dominant inherited diseases such as OPMD require a gene therapy approach that knocks down or inhibits the pathogenic protein (that is, PABPN1) and also has the ability to induce expression of a functional version of the same gene. Here we used an AAV-based gene therapy approach using a combination of DNA-directed RNA interference (ddRNAi) to knock down the endogenous wild type and

mutant forms of the PABPN1 allele, and gene addition to replace the transcript with a compensatory normal human codon-optimized mRNA resistant to RNA interference-induced cleavage. A similar pre-clinical approach has been used to restore the functionality of rhodopsin in photoreceptors of the animal model of Retinitis pigmentosa[48,49]. Here we show that this approach is very effective for OPMD, a disease model of aggregopathy, suggesting that a similar strategy could be used for common neurological gain-of-function aggregopathies such as Parkinson's, Alzheimer's or Huntington's diseases. The complexity of the central nervous system makes such neurological disorders very difficult diseases to treat, but successful gene therapies have now been reported with the development of integrating (LV) and non-integrating (AAV) vectors with adequate tropisms[46]. Regarding muscular dystrophies, AAV is currently the most promising viral vector for *in vivo* gene therapy applications, but the treatment of most muscular dystrophies has been hampered by the necessity to treat the whole musculature, which encompasses 40% of the body weight. OPMD is particularly suitable for such a gene therapy approach, as a very restricted number of muscles are clinically affected at the onset of the disease, making local intramuscular injections the most realistic approach in clinical settings. The doses and volume of

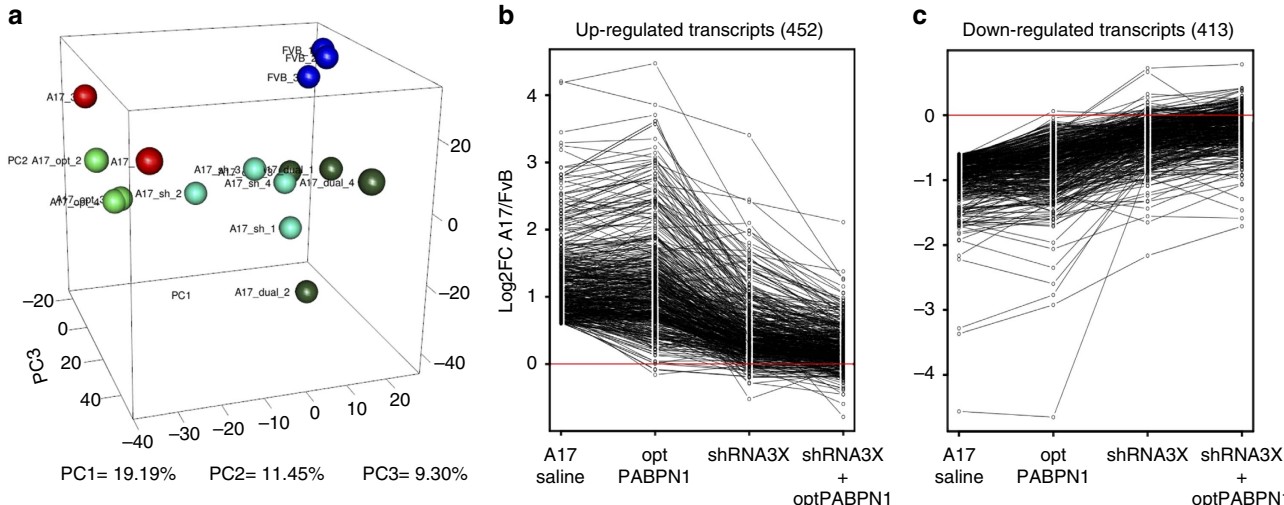

**Figure 5 | Transcriptome of A17 muscles treated with the combined approach is normalized to the one of WT mice.** (**a**) Principal component analysis of microarray data shows wide segregation of 30-week-old A17 mice clusters (in red) away from wild-type FvB mice clusters (in blue). Muscles treated with combined AAVs (shRNA3X and optPABPN1, in dark green) clustered very close to wild-type mice, while muscles treated with shRNA3X only (in light blue) clustered closer to FvB muscles than optPABPN1-injected muscles (light green). (**b,c**) Fold change in log2 (log2FC) of upregulated and downregulated genes in A17-treated groups compared to FvB. The treatments (A17 saline versus FvB, optPABPN1 versus FvB, shRNA3X versus FvB or shRNA3X + optPABPN1 versus FvB) are indicated under the graph. In A17 saline compared to FvB, 452 and 413 genes were up- and downregulated respectively. Ninety-eight per cent of these dysregulated (either up- or downregulated) transcripts showed a complete return to normal expression after the gene therapy treatment. $n = 3$ (A17 and FvB injected with saline and A17 injected with optPABPN1) or $n = 4$ (A17 injected with shRNA3X and A17 injected with shRNA3X + optPABPN1).

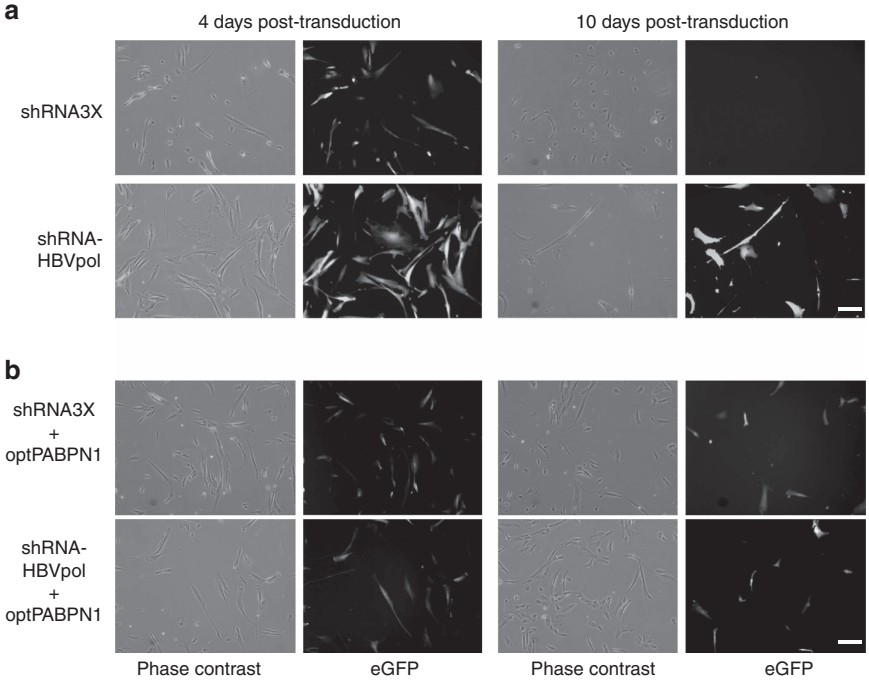

**Figure 6 | PABPN1 inhibition in human OPMD myoblasts induces cell death that can be prevented by optPABPN1 co-expression.** (**a,b**) Primary OPMD myoblast cells were transduced with shRNA3X tricistronic cassette, shRNA-HBVpol, shRNA3X-optPABPN1 or shRNA HBVpol-optPABPN1 LV vectors. GFP observed 4 and 10 days post transduction demonstrates that no GFP + cells survived on the condition transduced with shRNA3X. However, co-transduction with shRNA-HBVpol-optPABPN1 prevented cell death. Bar, 100 μm.

viral vectors we used in this study are in a standard range for intramuscular purpose in murine models, where a single injection is performed to transduce the whole TA muscle. In the context of potential human OPMD treatment, we envisage that AAV vectors for knockdown and replacement would be distributed in several points of the affected muscles by local intramuscular injection as

recently performed for autologous myoblasts (ClinicalTrials.gov NCT00773227).

It has previously been demonstrated in cultures of myoblasts that PABPN1 plays an essential role in their proliferation and differentiation[5]. PABPN1 is considered an essential gene involved in many biological pathways, and its expression is clearly crucial

for cell survival[50]. This was confirmed in *Drosophila*, where lethality was observed in *Pabp2* (homologue of human *PABPN1*) mutants and rescued with a *Pabp2* genomic transgene[6]. Our data in mouse and patient muscle cells confirm these observations, as PABPN1 depletion induced drastic muscle degeneration, myofibre turnover in mice and cell death in patient myoblasts. Interestingly, we also found that bringing a normal version of PABPN1 in affected mice and, more importantly, in human myogenic cells prevented the toxic effect of the PABPN1 knockdown, showing that lethal effects due to PABPN1 inhibition can be prevented and cell survival rescued. Furthermore, we clearly demonstrated that *in vivo* PABPN1 is required for muscle maintenance. Indeed, like in wild-type mice, in A17 mice the *in vivo* depletion of PABPN1 in skeletal muscle is detrimental and induces drastic muscle turnover. In these mice, the shRNA3X construct effectively silenced both normal and mutant endogenous PABPN1, and thus essentially depleted the muscle from all endogenous protein.

The difference in shRNA levels and PABPN1 expression we observed between muscles treated with shRNA3X only, and muscles treated with both shRNA3X and optPABPN1, likely reflects the turnover of the shRNA3X-expressing myofibres in the acute muscle degeneration/regeneration processes observed in the group treated with shRNA3X only compared with the groups of mice treated with both vectors (Fig. 3a–d). We propose that, in muscles treated with shRNA3X only, new endogenous satellite cells carrying the expPABPN1 activated in myoblasts proliferated and fused to replace the shRNA3X-expressing myofibres that were cleared. During muscle regeneration myoblasts express high-levels of PABPN1 (ref. 51), suggesting an essential role of PABPN1 during tissue repair. Muscles co-expressing shRNA3X and optPABPN1 were protected from the degeneration/ regeneration process (that is, from myofibre turnover), and so maintained higher level of shRNA molecules and lower levels of expPABPN1. Our data on the combined treatment suggest that the simple reduction of the protein and the consequent abrogation of insoluble nuclear aggregates are not sufficient to rescue the pathology, and that at least a small amount of normal protein needs to be expressed to allow cell survival and skeletal muscle stabilization.

This is consistent with published data showing that the steady-state level of PABPN1 (both protein and mRNA) is drastically lower in skeletal muscle compared to other tissues[51], which possibly explains why even if the genetic mutation is present body-wide only the skeletal muscle tissue manifests the clinical signatures of the pathology. Accordingly, the level of PABPN1 in normal muscles of FvB mice is almost undetectable[21] (Fig. 2b and Supplementary Fig. 1a).

It has been suggested that the presence of expPABPN1 nuclear aggregates induces in OPMD a depletion of available normal PABPN1. Therefore, the simple expression of a wild-type PABPN1 may be enough to compensate for the PABPN1 segregated into the intranuclear aggregates. Interestingly, in our study, the simple expression of normal PABPN1 alone was not sufficient to ameliorate the pathology in the A17 mouse model, though we cannot exclude that this was due to expPABPN1 being overexpressed in our mouse model. To the best of our knowledge, there is only one study showing the effect of wild-type PABPN1 overexpression in OPMD *in vivo* models: Rubinsztein and colleagues have shown that crossing a mouse overexpressing the wild-type bovine version of PABPN1 (that is, a repeat encoding including 10 alanine residues) with the A17 affected mouse ameliorates the dystrophic pathology, reducing muscle cell death due to apoptosis. In their study the amount of wild-type bovine PABPN1 likely matched the amount of bovine expPABPN1 in A17 mice[52]. However, the concurrent expression of the

endogenous wild-type murine version unbalanced the amount of PABPN1 towards the normal protein. This finding confirmed that exceeding the mutated PABPN1 with a normal version might potentially be enough in patient cells to see a small improvement. Surprisingly, we did not achieve such a substantial AAV-mediated overexpression *in vivo* of PABPN1, most likely because the system is tightly regulated due to feedback mechanisms that prevent the accumulation of a normal protein in cells[12], and only a small amount of functional protein can be expressed. Importantly, the expression of PABPN1 in muscles treated with both vectors likely derived only from the optPABPN1, as shown by MYC detection. Expression of optPABPN1 unbalanced the protein expression towards the normal PABPN1 version and this was sufficient to rescue the phenotype in the absence of expPABPN1. Our gene therapy treatment goes beyond the approach described by Rubinsztein and colleagues as we also abrogated the intranuclear inclusions (INIs), greatly ameliorating several aspects of the disease and essentially normalizing the transcriptome of the A17 mouse to the one of the FvB mice.

To summarize, while it is well known that a short alanine expansion in the PABPN1 protein leads to OPMD, why such mutation causes the pathology is still unknown[17]. One hypothesis is that the alanine expansion causes a relative loss of function by a decreased availability of normal PABPN1 at the protein level due, at least partially, to its sequestration in nuclear aggregates. Such loss of function is supported by the fact that decreased levels of PABPN1 result in defective myogenesis[51,53]. The second hypothesis is that the formation of nuclear aggregates induces a toxic gain of function by trapping both the normal PABPN1 and other molecules (proteins[54–58] and RNA[55,59–61]) crucial for several biological pathways. A pathological role of nuclear aggregates is supported by the effects of anti-aggregation therapies on improving OPMD phenotypes[45,62], and data showing that mutations in PABPN1 preventing aggregation do not lead to muscle pathology in *Drosophila* model[63]. The mouse model we used is characterized by a substantial overexpression of expPABPN1 that leads to the formation of aggregates (Fig. 7a,b). Our study shows that even by providing optPABPN1 there is no reduction in intranuclear aggregates, no pathology improvement and only a marginal change of muscle transcriptome (Fig. 7c). Furthermore, even abolishing the aggregates by inhibiting the endogenous PABPN1 results in no amelioration of the disease (Fig. 7d). Only the combined treatment rescues the normal PABPN1 expression and availability (Fig. 7e). These results indicate that PABPN1 aggregation induces in the murine OPMD model both a gain of function and a loss of physiological function. Interestingly, the fact that the transcriptome of shRNA3X-treated muscles is closer to the one of the FvB mice compared to the one of optPABPN1-treated muscles suggests that the pathology—at least in this mouse model—might be more dependent on a gain of function than on a loss of function of PABPN1. Notably, the changes in the transcriptome of AAV-shRNA3X treated muscles do not translate into functional improvement due to the substantial muscle damage.

Cell therapy is another appealing strategy to counteract OPMD, as the treatment could be performed locally, thus bypassing the issue of a systemic cell delivery. The local administration of autologous myoblasts in the cricopharyngeal muscle has been shown to improve swallowing function (ref. 28 and NCT00773227). While safe and associated to beneficial effects on a short term, autologous cells still harbour the mutation. An *ex vivo* gene therapy approach aiming to correct the genetic defect of myoblasts before the injection would be required to guarantee a long-lasting effect. We transduced cells derived from patient's biopsies using a lentiviral vector carrying

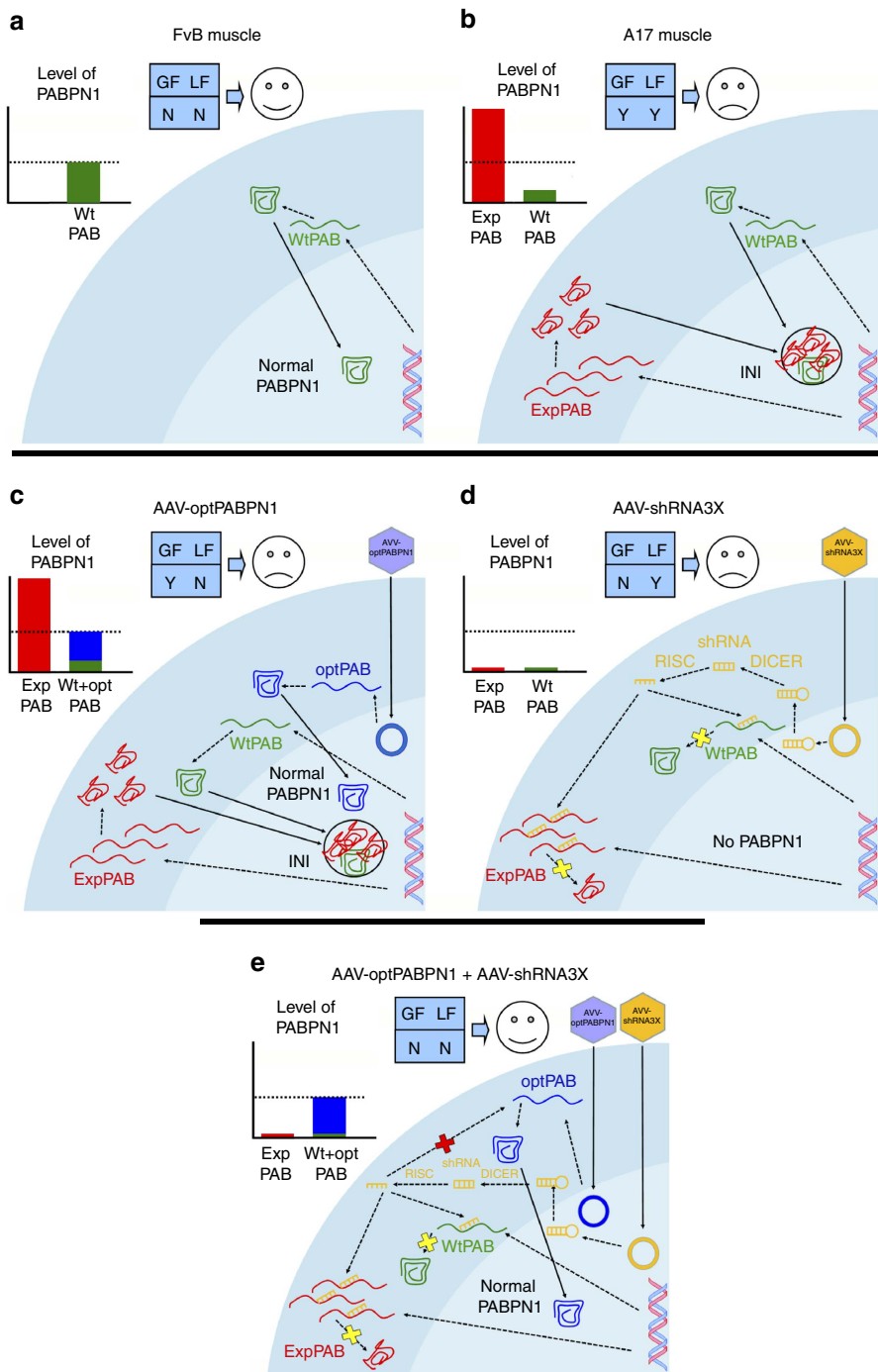

**Figure 7 | The combined treatment with AAVs abrogates INIs and restores normal PABPN1 expression improving both the gain of toxic function (GF) and the loss of physiological function (LF) mechanisms of the disease in the A17 mouse model.** (**a**) In wild-type FvB skeletal muscle, the endogenous wild-type PABPN1 (green) moves to the nucleus where it can function normally. The amount of normal PABPN1 that achieves the threshold for the optimal cell survival and functionality as shown in the graph in the upper left side. There is no loss of function (LF) or gain of function (GF), labelled with N (Not). (**b**) In A17 mice the expression of expPABPN1 (red) induces aggregate formation where normal PABPN1 and several other proteins as well as polyA RNAs are trapped. This might respectively induce both a loss of function (LF) associated to a minor amount of normal PABPN1 available and a toxic gain of function (GF) due to the sequestration of other macromolecules associated to dysregulated biological pathways, labelled as Y (Yes). (**c**) After injection of AAV-optPABPN1, the newly expressed optPABPN1 (blue) counteracts the lack of available normal PABPN1 (that is, it may reduce the loss of function mechanism of the disease) but it does not affect aggregate formation nor the toxic gain of function mechanism. No improvement of the pathology is observed. (**d**) By delivering ddRNAi to knockdown endogenous PABPN1, and after its conversion to siRNA by DICER and RISC complexes, aggregates are abrogated and this is likely associated to a reduction on the gain-of-function mechanism. However the normal PABPN1 downregulation promotes muscle degeneration. Again, no improvement of the pathology is observed. (**e**) Only the co-injection of the two vectors inhibits the aggregate formation while rescuing the normal PABPN1 to the level vital for optimal cell survival and functionality in A17 muscles (suggesting that both the toxic gain of function and the loss of function effect might be significantly reduced).

both the shRNA3X and the optPABPN1 cassettes on the same backbone, and we showed that the combined strategy was replicating at cellular levels the findings observed *in vivo* by AAV vectors.

The positive outcome in a relevant human model of disease suggests that the single bicistronic construct should also be tested for the AAV-gene therapy strategy, as it would greatly reduce the costs and efforts associated to the large-scale production of two AAV vectors. Furthermore, it paves the way for the clinical scenario where autologous myoblasts could be corrected *ex vivo* before transplantation.

In conclusion, our study suggests that OPMD might be due to both a gain of toxic function and a loss of function disease. We show for the first time that the *in vivo* gene therapy approach for the treatment of OPMD, a dominant disease model of aggregopathy, efficiently abolishes the formation of nuclear aggregates, normalizes muscle transcriptome and improves crucial hallmarks of the disease such as fibrosis and muscle atrophy, while increasing the muscle strength and normalizing the specific maximal force (that is, the muscle functionality) to that of unaffected muscles in a relevant animal model of the disease. These findings suggest that a similar approach could be used to treat other, more common, autosomal dominant gain-of-function neurological diseases and pave the way for the translation of this gene therapy strategy in clinical settings for OPMD.

## Methods

### Cell lines and cell transfection and transduction.
Human embryonic kidney cells (HEK293T, ATCC, Manassas, USA) were grown in Dulbecco's modified Eagle's medium (DMEM) containing 20 mM HEPES, 10% foetal bovine serum (FBS) and 2 mM glutamine (PAA laboratories, Yeovil, UK) in a humidified 5% $CO_2$ air atmosphere at 37 °C. HEK293T cell line was not authenticated. Cells were mycoplasma-free.

Human OPMD primary myoblasts isolated from a quadriceps muscle biopsy were obtained and provided by Myobank-AFM, France[64,65], which has an authorization from the Ministry and the CNIL (Authorization for sample disposal activity: No. AC-2013-1868). Muscle biopsy was obtained during surgical procedure after informed consent. Briefly, muscle biopsies were finely minced and explants were plated onto non-coated Petri dishes in drops of foetal calf serum (Invitrogen, Carlsbad, CA). Cells were cultured at 37 °C in a humid atmosphere containing 5% $CO_2$. Once mononucleated cells had migrated out from the explants, growth medium (that is, 199 Medium (Life Technologies) and DMEM (Life Technologies) in a 1:4 ratio) supplemented with 20% foetal calf serum, 5 ng ml$^{-1}$ human epithelial growth factor (Life Technologies), 0.5 ng ml$^{-1}$ bFGF, 0.2 μM dexamethasone (Sigma-Aldrich), 50 μg ml$^{-1}$ fetuin (Life Technologies) and 5 μg ml$^{-1}$ insulin (Life Technologies) was added. Cell populations were trypsinized when they reached 80% of confluence.

HEK293T cells were seeded at 300,000 cells per well. The day after, 1 h before transfection, the media was replaced with DMEM, 2% serum and no penicillin-streptomycin (PS). Cells were transfected with a solution of DMEM without serum and PS at a ratio of 125/1 DMEM/PEI (polyethyleneimine, linear MW 25,000) using DNA plasmids for pAAV-shRNA3X alone or in combination with pAAV-optPABPN1 or pAAV-expPABPN1. As a negative control, HEK293T cells were transfected with the plasmid pAAV-HBVpol, which expresses an shRNA against the HBVpol. Four micrograms per DNA plasmid was used per well. The HEK293T cells were incubated at 37 °C in DMEM with 2% FCS in the presence of antibiotics for 72 h. Cells were collected and lysed for analysis by western blotting.

Human myoblasts were seeded at 50,000 cells per well into six-well plates. The day after, cells were washed once with DMEM alone, and a 1 ml solution of DMEM, 10% foetal bovine serum and polybrene was added before transducing cells at MOI 100 with lentiviral vectors. The day after, medium was changed and cells were then cultured in growth medium.

### Generation of AAV and LV plasmid constructs and vectors.
Three shRNA, called sh-1, sh-2, sh-3, were initially designed to bind common sequences in human, mouse and bovine PABPN1. To generate a triple shRNA cassette (shRNA3X), these shRNAs were cloned downstream of the U61, U69 and H1 RNA polymerase III promoters, respectively. The shRNA3X construct was cloned into a self-complementary pAAV2 backbone to generate pscAAV-shRNA3X.

Human PABPN1 fused with a MYC-tag (optPABPN1) and a Kozak sequence was sequence-optimized by GeneArt (Waltham, USA). Both optPABPN1 and a mutant human expanded PABPN1 tagged with FLAG (expPABPN1) were subcloned into a ssAAV2 vector backbone downstream of SPc5-12 muscle-specific promoter[66] to generate pAAV-optPABPN1 and pAAV-expPABPN1. AAV vectors expressing shRNA3X or optPABPN1 (scAAV2/8-shRNA3X and ssAAV2/9-optPABPN1, respectively) were prepared with the standardized double transfection protocol. Briefly, HEK293T cells were seeded in roller bottles and cultured in DMEM at 37 °C and 5% $CO_2$. When 50% confluent, cells were transfected using PEI with either pAAV-shRNA3X or pAAV-optPABPN1 and, respectively, pAAV helper cap8 (pDP8.ape) or pAAV helper cap9 plasmids (pDP9rs), and cultured in roller bottles for 3 more days. Cells were lysed and recombinant pseudotyped AAV vector particles were purified by iodixanol (Sigma-Aldrich) step-gradient ultracentrifugation. The copy number of vector genomes was quantified by dot blot hybridization and qPCR. The titres of the AAV batches used for the *in vivo* application in this study were $1 \times 10^{12}$ vp per ml for scAAV2/8-shRNA3X and $5.2 \times 10^{12}$ vp per ml for ssAAV2/9-optPABPN1. For local injection in C57BL/6 mice control muscles were dosed with a single-strand AAV8-CMV-GFP (ssAAV8-GFP) prepared using the protocol described for the other AAV vectors. LV vectors expressing the same triple shRNA cassette (shRNA3X) driven by the same RNA polymerase III promoters (U61, U69 and H1) were produced by subcloning the expression cassette into a LV backbone together with a PGK-eGFP expression cassette. The optPABPN1 sequence was subcloned into a LV vector backbone where its expression was driven by SPc5-12 muscle-specific promoter (Supplementary Fig. 5). Unconcentrated and concentrated LVs were produced using HEK293T transfection[65] Briefly, HEK293T were cultured in DMEM high glucose-containing medium supplemented with 10% foetal calf serum (37 °C and 5% CO2). Lentiviral production was performed using a DNA/CaCl$_2$ transfection mix of a combination of the lentiviral vector, packaging plasmids and an envelope expressing plasmid. Forty-eight hours post transfection, the medium containing LVs was collected and spun down by centrifugation to remove cell debris. The supernatant was filtered (0.22 μm) to get a final clarified solution. The filtrate was transferred into ultracentrifuge tubes and centrifuged at 4 °C for 1 h 30 min at 22,000 r.p.m. Lentiviral pellets were suspended in PBS and left at 4 °C for at least 2 h (up to overnight). The final clear viral solution was aliquoted and stored at −80 °C until use. Titration was performed by qPCR and flux cytometry.

### In vivo experiments.
A17.1 transgenic mice (called here A17) express expanded (17 alanines) PABPN1 cDNA construct under the control of the human skeletal actin promoter and present progressive muscle weakness[21,24]. A17 mice and wild-type (WT) FvB controls were generated by crossing the heterozygous carrier strain A17 with the FvB background mice and genotyped by PCR for bovine PABPN1 4 weeks after birth. C57Bl6 mice were obtained from Janvier Labs. Animals were housed with food and water *ad libitum* in minimal disease facilities (Royal Holloway, University of London). Ethical and operational permission for *in vivo* experiments was granted by the RHUL Animal Welfare Committee and the UK Home Office, and this work was conducted under statutory Home Office regulatory, ethics, and licensing procedures, under the Animals (Scientific Procedures) Act 1986 (Project Licence 70/8271). All treated mice were used for the analyses. Briefly, twenty 10–12-week-old A17 male mice were randomized based on body weight and age and anaesthetized with isoflurane, and $2.5 \times 10^{10}$ vp of scAAV8-shRNA3X or $1.3 \times 10^{11}$ vp of ssAAV9-optPABPN1 or the two vectors in combination were diluted in 50 μl saline and intramuscularly administered into both TA muscles in five mice for each condition (that is, five mice per experimental group). Also, five of these A17 mice and five age-matched male FvB mice were injected with saline in both TA muscles as negative and healthy controls, respectively. Three 10–12-week-old C57BL6 male mice were injected in the TA muscles with $2.5 \times 10^{10}$ vp of scAAV8-shRNA3X or with $2.5 \times 10^{10}$ vp of ssAAV8-GFP in the contralateral TA as a control. At 18 weeks post injection, all mice were anaesthetized with pentobarbital, and *in situ* TA muscle physiology was performed. After analysis mice were weighed, killed and TA muscles were excised from tendon to tendon, weighed and frozen in liquid nitrogen-cooled isopentane.

### Muscle force measurement.
Mice were anaesthetized using a pentobarbital solution (intraperitoneally, 60 mg kg$^{-1}$) and contractile properties of TA muscles were analysed by *in situ* muscle electrophysiology[67]. A blind analysis was performed by the investigator. The knee and foot were fixed with clamps and the distal tendons of the muscles were attached to an isometric transducer (Harvard Bioscience) using a silk ligature. The sciatic nerves were proximally crushed and distally stimulated by a bipolar silver electrode using supramaximal square wave pulses of 0.1 ms duration. All data provided by the isometric transducer were recorded and analysed using the PowerLab system (4SP, AD Instruments). All isometric measurements were made at an initial length L0 (length at which maximal tension was obtained during the tetanus). Response to tetanic stimulation (pulse frequency: 75–150 Hz) was recorded and the maximal force was determined. For the measurement of muscle susceptibility to contraction-induced injury, muscles were stimulated at 150 Hz nine times. The stimulation-stretch cycle was repeated every 45 s. At each contraction, muscles were lengthened by 0.15 mm corresponding to 15% of muscle length. Maximum force was measured after each eccentric contraction and expressed as a percentage of the initial maximal force. After contractile measurements, muscles were collected, weighed to calculate the specific maximal force and frozen in isopentane cooled in liquid nitrogen and stored at −80 °C.

**qRT-PCR analysis.** Total RNA was extracted from skeletal muscles using Trizol (Invitrogen) according to the manufacturer's instructions. RNA quality and purity was determined using a ND-1000 NanoDrop spectrophotometer (NanoDrop Technologies) and RNA (50–250 ng for muscle biopsies, 1–3 µg for cell pellet) was reverse transcribed using M-MLV reverse transcriptase (Invitrogen) according to the manufacturer's instructions. cDNA was used for qPCR reaction using SYBR green mix buffer (LightCycler 480 Sybr green I Master) in a total of 9 µl reaction volume. PCR reactions were performed as follows: 8 min at 95 °C followed by 50 cycles: 15 s at 95 °C, 15 s at 60 °C and 15 s at 72 °C. The specificity of the PCR products was checked by melting curve analysis using the following programme: 65 °C increasing by $0.11\,°C\,s^{-1}$ to 97 °C. The expression level of each mRNA was normalized against murine *RPLP0* mRNA (large ribosomal protein, subunit P0) expression. Expression levels were calculated according to the ΔΔCt method. The sequences of primers used for RT-PCR and for qRT-PCR are included as Supplementary Note 1.

**Affymetrix mouse 2.0 ST arrays expression profiling.** Quality control was performed on total RNA extracted from mouse muscles using a Nanodrop and Bioanalyzer (Agilent Technologies, Santa Clara, CA). RNA quality and purity was determined using a ND-1000 NanoDrop spectrophotometer (NanoDrop Technologies) followed by Quality Control analysis (Agilent 2100 Bioanalyzer, Eukaryote Total RNA Nano assay, Agilent RNA 6000 Nano Kit, Agilent RNA Nano LabChip) to determine the RNA integrity number (RIN, 28S/18S ratio). Ninety nanograms of total RNA per sample was further processed using the GeneChip WT PLUS Reagent Kit for whole transcript expression profiling (Affymetrix, Santa Clara, CA). The sense strand ss cDNA was purified, fragmented and labelled using DNA-labelled reagent and then hybridized overnight onto the Mouse 2.0 ST arrays (100 or 81/4-Format). Prior to hybridization, the arrays were registered on GeneChip Command Console Software (AGCC) (Affymetrix, Santa Clara, CA). After the overnight hybridization, the arrays were washed and stained using the GeneChip Hybridization on a Fluidics 450 station (Affymetrix, Santa Clara, CA) and then scanned using the AGCC scan control software (Affymetrix, Santa Clara, CA). The raw data 'cel' files were analysed using Bioconductor (R software[68]). Raw signal intensities were quantile normalized with oligo package[68] and differential expression analysis was performed using the limma package[69]. Principal component analysis was performed using prcomp function and rgl package.

**Quantification of shRNA 1-3 copy number in muscles.** For shRNA copy number analysis, custom RT and quantitative PCR assays were developed for each hairpin. Standard curves used for absolute quantification were generated for each sequence utilizing a 10-fold dilution series ($10^8$–$10^2$ copies) of HPLC-purified RNA oligos (Sigma-Aldrich). Diluted RNA oligo standards and 50 ng of purified RNA from each TA muscle sample were reverse transcribed into cDNA using miScript II RT kit (Qiagen) according to manufacturers' instructions and a 2 µl aliquot of the five-fold diluted cDNA from the reverse transcription reaction was used as template in the subsequent qPCR assay (miScript SYBR Green PCR kit, Qiagen). qPCR reaction master mix and programme settings were followed according to manufacturers' instructions except for the use of a 4 µM forward primer specific to the hairpin sequence (OPMD hairpin primers are reported as Supplementary Note 1).

**Western blot.** Lysates from *in vitro* experiments were prepared by homogenizing cells in RIPA buffer containing: 0.15 M NaCl, 0.1% SDS, 50 mM Tris (pH 8), 2 mM EDTA and 10% Triton X–100 with protease inhibitor cocktail (Complete, Roche Diagnostics). Muscle lysates were prepared by homogenizing tissue in RIPA solution (NaCl 0.15 M, HEPES 0.05 M, NP-40 1%, sodium dehoxycholate 0.5%, SDS 0.10%, EDTA 0.01 M) with protease inhibitor cocktail. Proteins were separated on 4–12% Bis-Tris gel (Invitrogen) and transferred onto a nitrocellulose membrane (Hybond ECL membrane; Amersham Biosciences) for 1 h at 30 V constant. Membrane was blocked by incubation in 5% milk in 0.1 M PBS, 0.2% Tween-20 for 1 h at room temperature (r.t.). Membrane was stained with primary antibodies raised against PABPN1 (Abcam rabbit monoclonal, ab75855, 1/10,000, overnight (ON)), Vinculin (Sigma-Aldrich mouse monoclonal SAB4200080, 1/10,000, ON), GAPDH (Abcam mouse monoclonal ab9485, 1/2,500, ON), MYC-tag (Abcam rabbit polyclonal, ab9106, 1/5,000, ON) or FLAG (F3165 Sigma monoclonal, 1/2,000 ON). Membranes were then incubated with appropriate secondary antibodies conjugated either to the fluorochrome 800 (Licor, 1:10,000, 1 h r.t.) or to horseradish peroxidase . The Odyssey system (Licor) and the G:Box system (Syngene) were used to detect the signals from the membranes. All uncropped western blots included on the study are reported as Supplementary Fig. 7.

**Histology and immunofluorescence analysis.** Muscles were cross-sectioned at 10 µm thickness using a Bright OTF 5000 cryostat (Bright Instruments, Huntingdon, UK). The tissue sections were placed on coated glass slides (VWR, Lutterworth, UK) and stored at − 80 °C prior to use. The following antibodies were used for immunohistochemical staining: anti-PABPN1 (rabbit monoclonal, diluted 1:100, Abcam ab75855, ON, 4 °C), anti-neoMHC[70] (rabbit polyclonal, 1:50, 1 h r.t.), anti-Laminin (rat monoclonal; diluted 1:800; Sigma-Aldrich L0663, 1 h r.t.),

anti-fibronectin (rabbit polyclonal, Dako 0245, 1:500, 1 h r.t.) or anti-collagen VI (rabbit polyclonal, Abcam ab6588, 1:200, 1 h r.t.). Secondary antibodies were Alexa fluor (Molecular Probes) conjugated to 488 or 594 fluorochromes. For nuclear aggregates PABPN1 immunostaining, a published protocol[21,24] was modified to counterstain the sarcolemma with Laminin. Briefly, the slides were fixed in paraformaldehyde 4% and soluble proteins were removed by incubating with KCl buffer (1 M KCl, 30 mM HEPES, 65 mM PIPES, 10 mM EDTA, 2 mM MgCl$_2$, pH 6.9) for 1 h at r.t. Sections were then blocked for 1 h with 1% normal goat serum in 0.1 M PBS, 0.1% Triton X-100 and then incubated overnight at 4 °C with PABPN1 primary antibody diluted in the same buffer. Slides were then incubated with the anti-Laminin antibody (Sigma, rat monoclonal 1:800, 1 h r.t.). After washings, sections were incubated with fluorophore-conjugated secondary antibodies. Slides were mounted with mounting medium vector containing 4′,6-diamidino-2-phenylindole (DAPI; 3 µg ml$^{-1}$; Sigma-Aldrich Ltd.) to visualize nuclei. The percentage of the area stained for collagen VI or Sirius red in cryosections was measured by averaging the results of five randomly captured fields ( × 20 magnification, covering most of the section) using NIH ImageJ analysis software. The same programme was used to count the amount of INI in five randomly captured fields and the number of centrally nucleated fibres. All immmunofluorescence images were captured and digitized using identical parameters of exposure, saturation and gamma-levels between specimens. Standard protocols for haematoxylin and eosin (H&E) and Sirius red stainings were used to calculate the percentage of centrally nucleated fibres and to detect the area covered by fibrosis.

**Statistical analysis.** All data are presented as mean values ± standard error of the mean (mean ± s.e.m.) and all *n* values represent biological replicates. GraphPad Prism (version 4.0b; GraphPad Software, San Diego, CA, USA) was used for the analyses. The same software was used to test for normal distribution of data (Kolmogorov–Smirnov test) and to evaluate the variance between groups of data. Statistical analyses were performed using the Student *t*-test, the one-way analysis of variance with the Bonferroni *post-hoc* analysis or the chi-squared analysis. A sample-size power analysis was not performed. A difference was considered to be significant at $*P < 0.05$, $**P < 0.01$ or $***P < 0.001$.

**Data availability.** Microarray data have been deposited in NCBI's Gene Expression Omnibus and are accessible through GEO Series accession number GSE89996. All relevant data are available from the corresponding authors upon reasonable request.

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

## Acknowledgements

We thank the iVector ICM Vectorology Core facility for lentiviral productions (ICM, Brain and Spine Institute, F-75013, Paris, France) and MYOBANK-AFM from the Institute of Myology (Eurobiobank, code BB-0033-00012) for providing the human OPMD myoblasts. We thank Karuna Panchapakesan and Susan Knoblach from the Children's National Medical Center (Washington, USA) for the array expression profiling. We thank Ludovic Arandel, Fanny Roth and Naira Naouar for technical support. The authors acknowledge the OPMD patients for their collaboration. This work was supported by Benitec Biopharma LTD, the Centre National pour la Recherche Scientifique, the Association Française contre les Myopathies (Research Programs 15123 and 17110), the University Paris VI Pierre et Marie Curie, the Institut National de la Santé et de la Recherche Médicale, the Fondation de l'Avenir (project ET1-622). P.K. has a salary from the Ministere de l'Education Nationale de la Recherche et de Technologie.

## Author contributions

G.D., C.T. and A.M. conceived and designed the study. A.M., P.K., H.B., S.A.J., M.P.E., A.F., V.S. and G.C. performed the research and A.M., P.K., K.M., S.C.B., J.L.S.G., V.M., M.G., G.B.B., D.A.S., G.D. and C.T. analysed the research. A.M., P.K., G.D. and C.T. wrote the manuscript. All authors read and approved the final manuscript.

## Additional information

**Competing interests:** A patent named 'Reagents for treatment of oculopharyngeal muscular dystrophy (OPMD) and use thereof' has been filed by Benitec Biopharma and includes G.D. and C.T. as named inventors. D.A.S., V.S. and M.G. are employees of Benitec Biopharma. The other authors declare no conflicting financial interests.

