## [Peer review file · Nature Communications]

Reviewers' comments:

Reviewer #1 (Remarks to the Author):

This manuscript provides a wealth of original data that sheds some light on OPMD pathophysiology while being one of the first essays for combined viral treatment of a dominant aggregopathy in which the endogenous wildtype and mutated protein is down regulated and is replaced by a gene expressing a normal version of the protein. Considering the originality of the work, the quality of the multiple data sets and the potential for serving as the basis for the launch of a therapeutic trial I think it merits to be published in a high impact publication. This being said, it still lacks some editing to make its reading easier and some of the somewhat overinflated conclusions more conservative.

A. Summary of the key results

Strengths

First adenoviral trial in an OPMD mice model.

Confirms that by downregulation the expression of endogenous PABPN1 and expanded transgene PABPN1 combined to the transduction of a wildtype PABPN1 there can be a reversal of pathological phenotype at the tissue level and restoration of muscle strength in an overexpression transgenic PABPN1 model.

Similar strategy in OPMD patient myoblasts restores normal cellular properties.

Very strong experimental paper.

Overall good presentation of results.

Implications for possible treatment strategies for other dominant aggregopathies.

They present data that a single vector expressing the shRNA and the normal PABPN1 is the way to go, rather than using two vectors.

Weaknesses

Though they studied the most used "OPMD transgenic model", this massive overexpression model is in my opinion one with more limited implications for the human disease than usually publicized. In many ways, the authors confirm in their manuscript that by cutting down the level of expression significantly of the mutated transgene they contribute greatly to improving the pathology though with the known limitation that if you go too far you kill the cells since PABPN1 is essential. Sadly no other better transgenic model is available... It is their data on the human myoblasts that allow one to think that their strategy may be of value in the disease.

I really dislike the use of "healthy" protein throughout the text to talk about the transduced normal protein. It sounds too much like a salesman's expression, though I suspect it is more likely an imperfect translation of the French word: "saine". I would still use "normal" in most of the sentences.

Figure 7 and the discussion on gain and loss of function hypotheses are not very strong and do not add greatly to the paper. The paper itself in my opinion does not greatly contribute to settle the issue of gain or loss of function in OPMD, and since the transgenic mice used is a massive overexpression pro-gain of function model I'm not sure their results on the model can directly be transferred to the human pathology.

B. Data & methodology: validity of approach, quality of data, quality of presentation

High quality data overall.

C. Appropriate use of statistics and treatment of uncertainties

Except for Figure 5, all data is well presented. Figure 5: The legend to the figures, in particular 5B and 5C do not make the figure straight forward to interpret since there is no expression profile on wild type muscle to claim that there is a normalization.

D. Conclusions: robustness, validity, reliability

The conclusions are solid with the proviso of the limitation of this overexpression model as stated above.

E. Suggested improvements: experiments, data for possible revision

I have no new experiments to suggest, just some minor adjustment in the conclusions.

F. References: appropriate credit to previous work? Yes.

G. Clarity and context: lucidity of abstract/summary, appropriateness of abstract, introduction and conclusions.

I made many suggestions for minor changes as listed below.

Minor comments and suggestions

Be sure that PABPN1 for the gene is in italics through the text.

Line 48: add "cryptic" before trinucleotide

Line 49: Be sure that when referencing to the gene PABPN1 is in italics.

Line 50: "...aggregates that contain expanded PABPN1 and other proteins and RNAs". The sentence suggests that they contain only expPABPN1.

Line 50: The presence of excess fibrosis in patient OPMD muscle is not as clear as some would state such as the Netherlands' group and may be more the case in the A17 mice. Muscle fibrosis fiber atrophy is not an important part of OPMD pathology according to the classical work of Tomé and Fardeau.

Line 57: This last statement is too strong a statement. It should read: These results pave the way to gene replacement as an approach for OPMD human treatment.

Line 63: Would remove cricopharyngeal, since the cricopharyngeal are maybe even less affected than the other pharyngeal muscles and anyhow are part of the pharynx muscles.

Line 77: Move "in OPMD" to after: "a protective role".

Line 79: "interventional cure" does not mean anything specific as a category of treatment. Replace by "no medical treatment" or simple "cure" are available to arrest the disease.

Line 81: Please modify: "... degradation leading to an increased frequency of death...". Most patients do not die of complication of their OPMD but of other causes related to their advanced age.

Line 84: Not sure this is clear? Do they mean number of cells implanted influences response or something else?

Line 87: add: alpha actin promoter.

Line 90: Not everyone in the field think the mitochondrial changes are important or constant in the OPMD.

Line 100: state more like: small size of the expansion and important sequence homology.

Line 102-104: This is a misleading statement: "OPMD...required." Should read: "The dysphagia and ptosis of OPMD, with their limited number of muscle to treat...".

Line 115: Cannot say I like "aggregate disorder" label and would simply states that: Since wild type and mutated PABPN1 aggregate in OPMD.

Line 168: Replace "healthy" by "transduced normal PABPN1".

Line 178: "... was packaged as A singled...". Missing an "a".

Lines 180-181: Should write: "progressive muscle atrophy and weakness". Atrophy alone suggests a neuropathic etiology to fiber atrophy.

Line 201: need to qualify the nature of the difference: up or down?

Line 212: Is "clearance" the good word? It could only mean less accumulation because less is produced and the clearance stays at the same rate. Clearance would need to be assessed differently, and I suspect it would not be changed. I would change to: "... down-regulating PABPPN1 expression leads to a smaller number of aggregates". I would not qualify as "drastic" a change from 35% to 10%. Maybe: very important?

Lines 214-215: That is true, but how is its expression regulated? This will be a major challenge of this approach since we are talking of less significant baseline expression in a normal setting in humans than in these mice that overexpress the mutated PABPN1 to a very high level. In other words, a small drop or the impossibility to increase the expression of normal PABPN1 in cells when required could lead to cell death in patients' muscles.

Line 226: This cannot be a reasonable conclusion since the two muscles are so different in their PABPN1 expression. The relative baseline level of PABPN1 expression in A17 is 20-40X normal. How can they be compared?

Line 235: I do not think that this was ever proposed a rational way to treat the disease, so why make it a highlighted statement?

Line 263: I would use "normal" instead of "healthy".

Line 282: I would change "drastic improvement" which does not mean much and implies too much, to simply: a normalization of the muscle transcriptome.

Line 310: Would write: "These data confirm that inhibition of endogenous PABPN1 expression is ..."

Lines 315-319: Would remove "intranuclear" and change sentence to: ... cellular inclusions as observed in other so called "aggregopathies...".

Lines 317-321: Too long a sentence. Would cut at "dysphagia", remove "while" and make an independent sentence of the rest.

Lines 344- 346: This is true mostly at the beginning but it becomes a more generalized muscular dystrophy with time.

Lines 373-376: This sentence is not clear and seems to suggest that the data presented does help settle the issue of the toxicity of the inclusions, which I do not think is true. They show no data where the expanded protein is the only one that has its level diminished.

Lines 385-387: Again, since it is an overexpression model, I do not think this opinion stands.

Lines 420-425: This is true in this overexpression model but not necessarily in the human muscles.

Lines 441-442: Again the gain of function is only well document in overexpression models. In fact

overexpressing the wildtype may lead to aggregation formation though they are not as resistant to degradation.

Line 843: Change "healthy" to endogenous normal and expanded.

Line 859: Is it not better to use transduced than transfected for viruses?

Line 866: I think that it should be stated that FvB stands for the non-transgenic expression line.

Line 872: Not clear what the three blots of Figure 2B correspond to? Why 3, when they seem to show the same thing?

Line 888: Add: Inclusions are in green.

Line 914: "Prevents" is not the right word I think. You need to have a dynamic figure over time to claim that it prevents. I would simply use "diminishes".

Line 932: Figure 5. The legend to the figures, in particular 5B and 5C do not make the figure straight forward to interpret since there is no expression profile on wild type muscle to claim that there is a normalization.

Figure 7: This figure is hard to follow and I think tries too much to combine their data and the opposing gain and loss function hypotheses. I do not think their mice results help to settle the issue of which predominates in the human pathology, and their myoblast data does not in any way since no aggregation are observed in myoblasts.

Reviewer #2 (Remarks to the Author):

In this study, Dickson lab described an AAV gene therapy for OPMD. The disease is due to a relatively small trinucleotide repeat expansion which results in aggregation of PABPN1. The pathogenic mechanisms may likely involve a gain of toxic function due to pathologic protein aggregates and a loss of normal PABPN1 function. The authors applied RNA silencing and re-expression of a codon-optimized PABPN1 to the mouse model by local AAV injection and also in patient's cells. They achieved some histology and function improvement as well as transcriptome improvement. Similar strategies have been used to treat other diseases such as Retinitis Pigmentosa etc. But this is the first application in OPMD.

Major

1. I have a hard time to understand statistical markings in figures. The authors need to explicitly state which groups are being compared in figure legend for each marking. As it stands now, I cannot tell.
2. Figure 2B. They should include Myc western blot to distinguish from endogenous PABPN1.
3. Figure 2E. In figure legend, the authors stated that "The lower band represents the endogenous MYC expressed by muscle tissue.". I'm not aware any muscle that has an endogenous "MYC" tag.
4. Per Figure 4B, the myofiber cross-sectional area was reduced by almost half for A17 mice compared to that of FVB mice. However, from the images shown in Figures 2H and 3A, I cannot see any difference between A17 and FVB. It is also intriguing that the A17 image in Figure 4A shows a great variety in myofiber size. But in Figures 2H and 3A, A17 mouse muscle show quite uniform size. In light of this important differences, the author should (1) present full-view muscle cross-section images in Supp data, and (2) include serial sections for Figure 4A to show HE, laminin, DAPI and PABPN1 staining as well.
5. I'm not convinced by the data shown in Figure 3C. The authors should provide protein quantification too. Further, regeneration should be confirmed by embryonic myosin heavy chain staining.
6. Most of the fibrosis in mouse muscle is from collagen I and III. The authors performed immunostaining with collagen IV. Additional histochemical staining (such Sirius red, Masson trichrome) should be included. Further, considering mitigation of fibrosis is a major outcome, this should be quantified with hydroxyproline assay and/or western blot to show reduction of fibrosis in whole muscle (this will prevent bias due to sectioning and photo taking).
7. RNAi treatment (shRNA3x) alone has worsen the phenotype but the transcriptome profile is improved compared to that of optPABPN1 in Figure 5. This needs clarification.

Minor

1. The vector dose used in mouse study needs justification and it will be nice if authors can briefly discuss applicability of this dose in human patients.

2. Several reference citations are incomplete such 26 and 31.
3. The statement "the pathology might be more dependent from a gain of function than from a loss of function" is speculative and should be removed.
4. When measuring muscle contractility in muscular dystrophy mouse studies, response to eccentric contraction is often included as a sensitive assay to detect the damage and rescue of function. Is there a reason for not doing this assay?
5. Figure 1D western should include Flag blot to illustrate expPABPN1 expression.
6. Figure 2H bottom panel images are barely visible.
7. The images used in the figure is of poor quality. Suggest to include large photos in Supplements.
8. In Supp data, they provided shRNA3X construct sequence. It will be very helpful if the authors can mark the location of promoters and shRNA sequence use different color or different font etc.

Reviewer #3 (Remarks to the Author):

The manuscript describes gene therapy in a mouse model of oculopharyngeal muscular dystrophy. The disease is a protein aggregation disease and surgery is currently the main treatment but is inadequate and patients die of complications of failure to swallow properly. Anti-aggregation small molecules and intrabodies are at various stages of testing but treatment targeting the genetic defect may provide permanent treatment. The authors have designed a two-vector in vivo strategy in the mice – to knock down the mutant expanded protein with one vector while introducing a knock-down resistant non-expanded gene. They do so using a tricistronic silencing vector alongside a gene supplementation vector. They show by various measures that local muscle pathology in the mouse model is improved by intramuscular injection of vector into the tibialis anterior. Commendably, they also demonstrate no toxic effect of injection of these vectors into wild type mice – toxicity that might arise, e.g., from full knock-down & insufficient correction, or from saturation of the molecular machinery involved in siRNA activity. Also, commendably, they test their system in human cells. There are two specific, major concerns, which should be addressed.

- 1) No analysis of ectopic expression – both sides of the coin. Did the authors look for evidence of correction of pathology in distal muscles? More importantly, did the authors look for vector spread beyond the site of injection? Finally, and even more importantly, did the authors look at what happened when they intentionally delivered the vector non-locally i.e. by systemic injection?
- 2) How would one administer to the relevant muscles in humans? The authors allude to this, but do not detail how it might be achieved. Has this been performed with any substance, in humans or large animals, previously?
- 3) Related to point (2) – are the authors sure that there are only very restricted muscles affected, or are they the only ones that kill the patient first? What happens when those are treated? Perhaps then other aggregations would occur in other muscles. Has anyone ever looked at post-mortem?

Minor comments:

Line 124/125 “the one” should probably read “that”

Line 538 “intraperitoneally”...

Figure 7 is really hard to understand – unfortunately a case of a picture not painting 1000 words. Use full words, describe more clearly on each part of the figure – maybe a brief narrative under each part/panel

**Author response to Reviewer #1 Comments: Manuscript NCOMMS-16-16130, Malerba et al
PABPN1 gene therapy rescues disease phenotype in oculopharyngeal muscular dystrophy**
[NB: Author responses are in italics throughout]

We thank the reviewer who seemed to have really appreciated our work and stated that our manuscript “merits to be published in a high impact publication”. His/her concerns are addressed below.

This manuscript provides a wealth of original data that sheds some light on OPMD pathophysiology while being one of the first essay for combined viral treatment of a dominant aggregopathy in which the endogenous wildtype and mutated protein is down regulated and is replaced by a gene expressing a normal version of the protein. Considering the originality of the work, the quality of the multiple data sets and the potential for serving as the basis for the launch a therapeutic trial I think it merits to be published in a high impact publication. This being said, it still lacks some editing to make its reading easier and some of the somewhat overinflated conclusions more conservative.

A. Summary of the key results

Strengths

First adenoviral trial in an OPMD mice model.

Confirms that by downregulation the expression of endogenous PABPN1 and expanded transgene PABPN1 combined to the transduction a wildtype PABPN1 there can be a reversal of pathological phenotype at the tissue level and restoration of muscle strength in an overexpression transgenic PABPN1 model.

Similar strategy in OPMD patient myoblasts restores normal cellular properties.

Very strong experimental paper.

Overall good presentation of results.

Implications for possible treatment strategies for other dominant aggregopathies.

They present data that a single vector expressing the shRNA and the normal PABPN1 is the way to go, rather than using two vectors.

Weaknesses

Though they studied the most used “OPMD transgenic model”, this massive overexpression model is in my opinion one with more limited implications for the human disease than usually publicized. In many ways, the authors confirm in their manuscript that by cutting down the level of expression significantly of the mutated transgene they contribute greatly to improving the pathology though with the known limitation that if you go too far you kill the cells since PABPN1 is essential. Sadly no other better transgenic model is available... It is their data on the human myoblasts that allow one to think that their strategy may be of value in the disease.

Response: *as the reviewer suggested, the A17 mouse, where the pathology is due to expPABPN1 overexpression, is not a perfect model of OPMD but, as he correctly mentioned, this is the best mammalian model we can use. We previously demonstrated that this mouse model shares most of the features of the human OPMD (Anvar et al, 2011; Chartier et al, 2015). The mouse model that best reproduces the genetic defect of the human mutation (made by expressing 13 alanine expanded PABPN1 under the control of its endogenous promoter) shows actually only a neurological phenotype and no myopathy indicating the need of expPABPN1 overexpression to reproduce the muscle pathology in mouse (Dion et al, 2005). To strengthen our conclusions we included experiments in muscles of mice expressing normal PABPN1 to show a detrimental effect of PABPN1 depletion and in human OPMD myoblasts*

where the effect of the gene therapy treatment is expected to best mimic the possible outcome in human.

I really dislike the use of “healthy” protein throughout the text to talk about the transduced normal protein. It sounds too much like a salesman’s expression, though I suspect it is more likely an imperfect translation of French word: “saine”. I would still use “normal” in most of the sentences.

Response: we revised the manuscript and used “normal” instead of “healthy” as suggested by the reviewer.

Figure 7 and the discussion on gain and loss of function hypotheses are not very strong and do not add greatly to the paper. The paper itself in my opinion does not greatly contribute to settle the issue of gain of loss of function in OPMD, and since the transgenic mice used is a massive overexpression pro-gain of function model I’m not sure their results on the model can directly be transferred to the human pathology.

Response: we understand the point of the reviewer as the mouse model we used is made by substantially overexpressing *expPABPN1*, which is what induces aggregate formation (and then both the gain of toxic function and some of the observed loss of function). However the only relevant difference is that in human one allele is mutated and 50% of *expPABPN1* is sufficient to drive the phenotype while in the mouse model the overexpression of a newly bovine *expPABPN1* inserted gene induces the pathological effect. This means that while the loss of function in human is likely due to a combination of a reduced expression of normal *PABPN1* (i.e. 50% reduction in heterozygous patients) and the sequestration of normal *PABPN1* trapped in aggregates, in the A17 mouse model the loss of function only originates from the sequestration of normal *PABPN1* into the aggregates as both the endogenous murine alleles expressing normal *PABPN1* are functional. We made some adjustments to the introduction (line 136) and discussion (lines 451, 458, 461, 482). Furthermore we modified Figure 7 to adjust the explanation of the possible mechanisms originating the pathology to our mouse model (and not necessarily to the human disease) and we wrote a new Figure legend 7 to make our conclusions more conservative.

B. Data & methodology: validity of approach, quality of data, quality of presentation
High quality data overall.

C. Appropriate use of statistics and treatment of uncertainties

Except for Figure 5, all data is well presented. Figure 5: The legend to the figures, in particular 5B and 5C do not make the figure straight forward to interpret since there is no expression profile on wild type muscle to claim that there is a normalization.

D. Conclusions: robustness, validity, reliability

The conclusions are solid with the proviso of the limitation of this overexpression model as stated above.

E. Suggested improvements: experiments, data for possible revision

I have no new experiments to suggest, just some minor adjustment in the conclusions.

F. References: appropriate credit to previous work? Yes.

G. Clarity and context: lucidity of abstract/summary, appropriateness of abstract, introduction and conclusions.

I made many suggestions for minor changes as listed below.

Minor comments and suggestions

Response: we greatly thank the reviewer for suggesting many valuable changes to the text

that improved the general readiness of the manuscript. We revised the manuscript accordingly. The page numbers refer to the version of the manuscript where all changes are tracked.

Be sure that PABPN1 for the gene is in italics through the text.

Response: *the text has been modified accordingly.*

Line 48: add "cryptic" before trinucleotide

Response: *We appreciate the suggestion of the reviewer as the number of GCG triplets on the expanded PABPN1 is variable between patients. However we decided not to add "cryptic" as this definition is more related to a phenotype depending on the presence of unexpected mutations more than on the type of mutation.*

Line 49: Be sure that when referencing to the gene PABPN1 is in italics.

Response: *the text has been modified accordingly (line 50).*

Line 50: "...aggregates that contain expanded PABPN1 and other proteins and RNAs". The sentence suggests that they contain only expPABPN1.

Line 50: The presence of excess fibrosis in patient OPMD muscle is not as clear as some would state such as the Netherlands' group and may be more the case in the A17 mice. Muscle fibrosis fiber atrophy is not an important part of OPMD pathology according to the classical work of Tomé and Fardeau.

Response: *we modified these sentences of the abstract that now reads "OPMD is characterized by the presence of nuclear aggregates. Affected muscles present fibrosis and muscle weakness." (line 53).*

Line 57: This last statement is too strong a statement. It should read: These results pave the way to gene replacement as an approach for OPMD human treatment.

Response: *the paragraph has been modified and it now reads "These results pave the way towards a gene replacement approach to OPMD treatment" (line 59).*

Line 63: Would remove cricopharyngeal, since the cricopharyngeal are maybe even less affected than the other pharyngeal muscles and anyhow are part of the pharynx muscles.

Response: *the text has been modified as suggested (line 67).*

Line 77: Move "in OPMD" to after: "a protective role".

Response: *the text has been modified as suggested (line 81).*

Line 79: "interventional cure" does not mean anything specific as a category of treatment. Replace by "no medical treatment" or simple "cure" are available to arrest the disease.

Response: *the text has been modified as suggested (line 83).*

Line 81: Please modify: "... degradation leading to an increased frequency of death...". Most patients do not die of complication of their OPMD but of other causes related to their advanced age.

Response: *the text has been modified and now reads: "...degradation leading to severe swallowing impairment, pulmonary infections and choking (line 85).*

Line 84: Not sure this is clear? Do they mean number of cells implanted influences response or something else?

Response: we clarified the sentence adding "...improved the pathology of OPMD patients with a cell dose-dependent improvement in swallowing (Perie et al, 2014) (line 89).

Line 87: add: alpha actin promoter.

Response: the text has been modified as suggested (line 93).

Line 90: Not everyone in the field think the mitochondrial changes are important or constant in the OPMD.

Response: we recently published a proteomic study performed on muscle biopsies from OPMD patients demonstrating mitochondrial defects (Chartier et al, 2015). This was also confirmed in the A17 mouse model. For clarity we have now added this reference mentioning that the A17 model recapitulates most of the features of OPMD patients, and removed the sentence related to mitochondrial changes (line 96).

Line 100: state more like: small size of the expansion and important sequence homology.

Response: the text has been modified as suggested (line 106).

Line 102-104: This is a misleading statement: "OPMD...required." Should read: " The dysphagia and ptosis of OPMD, with their limited number of muscle to treat...".

Response: the text has been modified as suggested (line 109).

Line 115: Cannot say I like "aggregate disorder" label and would simply states that: Since wild type and mutated PABPN1 aggregate in OPMD.

Response: the text has been modified and now reads: " Since expanded PABPN1 aggregates in OPMD..." (line 122).

Line 168: Replace "healthy" by "transduced normal PABPN1".

Response: the text has been modified as suggested (line 135).

Line 178: "... was packaged as A singled...". Missing an "a".

Response: the text has been modified as suggested (line 190).

Lines 180-181: Should write: "progressive muscle atrophy and weakness". Atrophy alone suggests a neuropathic etiology to fiber atrophy.

Response: the text has been modified as suggested (line 193).

Line 201: need to qualify the nature of the difference: up or down?

Response: the text has been modified adding "with a lower amount in the shRNA3X treated group..." (line 216).

Line 212: Is "clearance" the good word? It could only mean less accumulation because less is produced and the clearance stays at the same rate. Clearance would need to be assessed differently, and I suspect it would not be changed. I would change to: "... down-regulating PABPN1 expression leads to a smaller number of aggregates". I would not qualify as "drastic" a change from 35% to 10%. Maybe: very important?

Response: we modified the sentence that now reads "shRNA3X delivered by AAV efficiently down-regulates PABPN1 in vivo and significantly decreases the amount of insoluble PABPN1 aggregates." (line 227).

Lines 214-215: That is true, but how is its expression regulated? This will be a major challenge

of this approach since we are talking of less significant baseline expression in a normal setting in humans than in these mice that overexpress the mutated PABPN1 to a very high level. In other words, a small drop or the impossibility to increase the expression of normal PABPN1 in cells when required could lead to cell death in patients' muscles.

Response: *we agree with the reviewer that expressing optPABPN1 is crucial in cells where the endogenous PABPN1 is downregulated. However in human OPMD context, inducing a strong endogenous PABPN1 downregulation while achieving the threshold of optPABPN1 needed for the cell's life should be even easier than in a model where expPABPN1 is overexpressed. Indeed as mentioned by the reviewer, the amount of endogenous PABPN1 expressed in muscle is very low so the vector does not necessarily need to express very high amount of optPABPN1. The amount of protein we expressed with AAV-spc512-optPABPN1 or the same cassette cloned in LV was able to efficiently rescue cell survival after almost complete PABPN1 knock down mediated by the shRNA3X both in the mouse model and in human myoblasts. However this system can be improved further and possibly finely tuned: we are currently trying other configurations as single backbones including both cassettes, different promoters to express optPABPN1 (e.g. Muscle Creatine Kinase) and different modified shRNA3X cassettes to provide a variable level of PABPN1 downregulation.*

Line 226: This cannot be a reasonable conclusion since the two muscles are so different in their PABPN1 expression. The relative baseline level of PABPN1 expression in A17 is 20-40X normal. How can they be compared?

Response: *we modified the sentence in the manuscript to clarify that we are not comparing the normal and OPMD mouse models but just showing the effect of PABPN1 depletion in a muscle expressing only normal PABPN1 (line 244).*

Line 235: I do not think that this was ever proposed a rational way to treat the disease, so why make it a highlighted statement?

Response: *we modified the sentence in the manuscript (line 238).*

Line 263: I would use "normal" instead of "healthy".

Response: *the text has been modified as suggested (line 285).*

Line 282: I would change "drastic improvement" which does not mean much and implies too much, to simply: a normalization of the muscle transcriptome.

Response: *the text has been modified as suggested (line 304).*

Line 310: Would write: "These data confirm that inhibition of endogenous PABPN1 expression is ..."

Response: *the text has been modified as suggested (line 331).*

Lines 315-319: Would remove "intranuclear" and change sentence to: ... cellular inclusions as observed in other so called "aggregopathies... " .

Response: *the text has been modified as suggested (line 338).*

Lines 317-321: Too long a sentence. Would cut at "dysphagia", remove "while" and make an independent sentence of the rest.

Response: *the text has been modified as suggested (line 342).*

Lines 344- 346: This is true mostly at the beginning but it becomes a more generalized muscular dystrophy with time.

Response: we modified the sentence in the manuscript (line 369).

Lines 373-376: This sentence is not clear and seems to suggest that the data presented does help settle the issue of the toxicity of the inclusions, which I do not think is true. They show no data where the expanded protein is the only one that has its level diminished.

Response: in the A17 mouse model the intranuclear inclusions originate from expPABPN1 overexpression. As suggested by the reviewer we do not know if this is a cause or just an effect of the disease and indeed we do not suggest one mechanism to prevail on the other. We state that the solely reduction of aggregates by inhibition of expPABPN1 (which is the protein making the aggregates in the A17 mouse model) is not sufficient to improve the pathology. This is interesting as the use of anti-aggregation drugs or intrabodies proved to be effective by reducing aggregates (without knocking down expPABPN1) and improving the pathology in in vivo models of OPMD (see references 30-32 of the manuscript). To clarify the sentence we changed from “Our data on the combined treatment confirm that the simple reduction of the pathological protein...” to “Our data on the combined treatment suggest that the simple reduction of the protein...” (line 402).

Lines 385-387: Again, since it is an overexpression model, I do not think this opinion stands.

Response: we added a sentence to acknowledge the issue of the mouse model overexpressing expPABPN1 (line 416).

Lines 420-425: This is true in this overexpression model but not necessarily in the human muscles.

Response: we modified the text to specify that these data refer to the mouse model (line 461).

Lines 441-442: Again the gain of function is only well document in overexpression models. In fact overexpressing the wildtype may lead to aggregation formation though they are not as resistant to degradation.

Response: we modified the sentence to reflect the issue of working with an overexpression model. The sentence now reads “our study suggests that OPMD might be due to both a gain of toxic function and a loss of function disease” (line 482).

Line 843: Change "healthy" to endogenous normal and expanded.

Response: the text has been modified as suggested (line 946).

Line 859: Is it not better to use transduced than transfected for viruses?

Response: in this experiment plasmids were transfected using PEI. In vitro, only OPMD myoblasts were transduced using viral vectors (i.e. Lentiviral vectors)

Line 866: I think that it should be stated that FvB stands for the non-transgenic expression line.

Response: this information about FvB is already on the Materials and Methods section. We also specified this in the result section (line 198).

Line 872: Not clear what the three blots of Figure 2B correspond to? Why 3, when they seem to show the same thing?

Response: in figure 2 all the muscles analysed by western blot are reported with the aim of showing that results are consistent in all treated muscles. We have now simplified the figure 2 showing only one of the three immunoblots. The remaining two were moved to the Supplementary figure 1.

Line 888: Add: Inclusions are in green.

Response: *the text has been modified as suggested (line 995).*

Line 914: "Prevents" is not the right word I think. You need to have a dynamic figure over time to claim that it prevents. I would simply use "diminishes".

Response: *we corrected the text as suggested (line 1031).*

Line 932: Figure 5. The legend to the figures, in particular 5B and 5C do not make the figure straight forward to interpret since there is no expression profile on wild type muscle to claim that there is a normalization.

Response: *we modified the legend of figure 5 (page 49) to make the interpretation of the results easier. Please note that there is no expression profile of wild type muscles because all data are normalized to them.*

Figure 7: This figure is hard to follow and I think tries too much to combine their data and the opposing gain and loss function hypotheses. I do not think their mice results help to settle the issue of which predominates in the human pathology, and their myoblast data does not in any way since no aggregation are observed in myoblasts.

Response: *As reported in one of the previous responses we modified the text to make more conservative our conclusions about the gain and loss function hypotheses. In particular we made some adjustments to the introduction (line 136) and discussion (lines 451, 458, 461, 482). Figure 7 has been modified to adjust the explanation of the possible mechanisms originating the pathology to our mouse model (and not necessarily to the human disease) and we wrote a new Figure legend 7 to make these conclusions more conservative.*

Author response to Reviewer #2 Comments: Manuscript NCOMMS-16-16130, Malerba et al, PABPN1 gene therapy rescues disease phenotype in oculopharyngeal muscular dystrophy
[NB: Author responses are in italics throughout]

We thank the reviewer for the careful revision of our manuscript. His/her criticisms are addressed below.

In this study, Dickson lab described an AAV gene therapy for OPMD. The disease is due to a relatively small trinucleotide repeat expansion which results in aggregation of PABPN1. The pathogenic mechanisms may likely involve a gain of toxic function due to pathologic protein aggregates and a loss of normal PABPN1 function. The authors applied RNA silencing and re-expression of a codon-optimized PABPN1 to the mouse model by local AAV injection and also in patient's cells. They achieved some histology and function improvement as well as transcriptome improvement. Similar strategies have been used to treat other diseases such as Retinitis Pigmentosa etc. But this is the first application in OPMD.

Major

1. I have a hard time to understand statistical markings in figures. The authors need to explicitly state which groups are being compared in figure legend for each marking. As it stands now, I cannot tell.

***Response:** The groups of compared mice are shown using a black line below the asterisks which number indicates how much relevant is the difference (* $p < 0.05$, ** $p < 0.01$, *** $p < 0.001$, ns: not significant). We believe this is a pretty standard and easy way to visualize which groups of samples are compared on the graphs. Following the reviewer's comment we further improved the legends to describe which groups of mice are compared.*

2. Figure 2B. They should include Myc western blot to distinguish from endogenous PABPN1.

***Response:** An example of western blot used for quantification of myc is indeed reported in figure 2E. This immunoblot shows that no difference was detected in Myc expression between muscles treated with optPABPN1 and muscles treated with shRNA3X and optPABPN1. Based on the reviewer's suggestion, we have now added in Supplementary Figure 1b the Myc-tag immunoblot showing the protein detection in the samples not included in figure 2E.*

3. Figure 2E. In figure legend, the authors stated that "The lower band represents the endogenous MYC expressed by muscle tissue.". I'm not aware any muscle that has an endogenous "MYC" tag.

***Response:** we never mentioned the presence of a Myc-tag in the muscle. Myc and other very similar members of its family are endogenously expressed in postdevelopmental skeletal muscle at early or late stages (Veal et al, 1998; Conacci-Sorrell et al, 2010) and they could be detected by the antibody we used as their molecular weight corresponds to the extra-band we detected in our western blot. However we repeated a WB hybridizing the membrane with only the secondary antibody and we detected an unspecific band at the same molecular weight as the band we mentioned as endogenous myc. While the specificity of the primary antibody for the Myc-tag is obvious, we cannot exclude that the lower band is actually due to unspecific binding of the secondary antibody and therefore we deleted the sentence "The lower band represents the endogenous MYC expressed by muscle tissue" from the legend.*

4. Per Figure 4B, the myofiber cross-sectional area was reduced by almost half for A17 mice compared to that of FVB mice. However, from the images shown in Figures 2H and 3A, I

cannot see any difference between A17 and FVB. It is also intriguing that the A17 image in Figure 4A shows a great variety in myofiber size. But in Figures 2H and 3A, A17 mouse muscle shows quite uniform size. In light of these important differences, the author should (1) present full-view muscle cross-section images in Supp data, and (2) include serial sections for Figure 4A to show HE, laminin, DAPI and PABPN1 staining as well.

Response: *the reviewer correctly states that some of the pictures chosen in figures 2-4 for A17/FvB did not consistently show a similar result in terms of myofibre cross sectional area. In Tibialis anterior muscle there is variability in myofibre size with the inner region (close to the bone) having fibres with smaller cross sectional area of the outer region (close to the skin). This is clearly visible on the Supplementary figure 3A showing the full-view muscle cross sections images that we have now included following the reviewer's suggestion. In order to be consistent with the CSA data, we changed some of the pictures in figures 2G (FvB saline) and 3 (FvB saline). We also changed the pictures for A17 saline and optPABPN1 relative to the set of collagen staining. These pictures are now included in the Supplementary figure 2. On the same figure we added the relative fields from serial sections stained with H&E and laminin/DAPI as suggested by the reviewer. We did not include PABPN1 staining as in our opinion this does not add any value to the figure. We instead added 2 more sets of pictures from serial muscle sections stained for sirius red and fibronectin as described below (see response to major point 6).*

5. I'm not convinced by the data shown in Figure 3C. The authors should provide protein quantification too. Further, regeneration should be confirmed by embryonic myosin heavy chain staining.

Response: *the muscle damage due to PABPN1 depletion is now clearly visible also in the cross muscle section of the muscle treated with shRNA3X only vector (Supplementary figure 3a). As suggested by the reviewer we also stained muscles with neonatal MHC antibody (Butler-Browne et al, 1982) and included in Figure 3 some representative pictures that clearly show a substantial amount of small regenerating muscle fibres (in green) only after injection of AAV-shRNA3X.*

6. Most of the fibrosis in mouse muscle is from collagen I and III. The authors performed immunostaining with collagen IV. Additional histochemical staining (such Sirius red, Masson trichrome) should be included. Further, considering mitigation of fibrosis is a major outcome, this should be quantified with hydroxyproline assay and/or western blot to show reduction of fibrosis in whole muscle (this will prevent bias due to sectioning and photo taking).

Response: *we did not use Collagen IV for our analyses, as indeed this is not a good marker of muscle fibrosis. We used Collagen VI that has been already included in many studies to measure fibrosis in skeletal muscle (e.g. Boldrin et al, 2009; Gibertini et al 2014). We agree with the reviewer that reduction in fibrosis is an interesting finding and deserves to be described better. Therefore we performed and quantified sirius red histochemical staining (as suggested by the reviewer) which showed a similar pattern to that observed for collagen VI. Additionally we performed immunostainings for collagen I and fibronectin that showed a similar trend with reduction of the proteins after treatment with a combination of AAV-shRNA3X and AAV-optPABPN1 vectors. We believe our quantifications of sirius red and collagen VI expression (further supported by stainings performed for other markers of fibrosis) are now sufficient to demonstrate a reduction in fibrosis in shRNA3X+AAV-optPABPN1 treated muscles compared with saline injected A17 muscles. By taking 5-6 fields (20X) per sample we covered almost all the section area so we think our analysis is not biased by intrinsic variability in the muscle sections. Notably this is an established method to analyze fibrosis in muscle as demonstrated by several studies we and others previously published (e.g. Malerba*

et al 2011; De Greef et al 2016). Quantification of sirius red and collagen VI has been included in Figure 4 while pictures taken from serial sections stained for sirius red, collagen I and fibronectin are now included on the panel showing H&E, collagen VI and laminin/DAPI in Supplementary figure 2.

7. RNAi treatment (shRNA3x) alone has worsen the phenotype but the transcriptome profile is improved compared to that of optPABPN1 in Figure 5. This needs clarification. **Response:** *the transcriptome analysis shows which genes are de-regulated in A17 treated with saline or with AAV vectors compared to FvB muscles and it is a good indication that biological pathways are improved/restored after a treatment. However this does not mean that A17 treated muscles with a transcriptome close to the one of FvB muscles are necessarily improved. Other aspects of the pathology have to be analysed and a comprehensive set of outcome measures must be evaluated to conclude that a treatment ameliorated the disease. The solely AAV-shRNA3X administration did not improve the phenotype (fibrosis seems to be actually enhanced in muscles treated with only this vector while there is no difference in muscle weight, CSA and muscle strength compared to A17 injected with saline). Furthermore there is a massive muscle degeneration/regeneration process ongoing that is definitely detrimental for the muscle. On the other hand the inhibition of expPABPN1 (and the consequent reduction of aggregates) provided by this vector is likely beneficial in terms of re-regulation of many genes. Our results suggest that the partial transcriptome re-regulation, after AAV-shRNA3X administration, does not translate in functional improvement due to the occurring muscle damage. A sentence has been added to the discussion to clarify these points (line 464, manuscript with tracked changes).*

Minor

1. The vector dose used in mouse study needs justification and it will be nice if authors can briefly discuss applicability of this dose in human patients.

Response: *we added a comment to the text (lines 370, manuscript with tracked changes).*

2. Several reference citations are incomplete such 26 and 31.

Response: *all incomplete references have been fixed.*

3. The statement "the pathology might be more dependent from a gain of function than from a loss of function" is speculative and should be removed.

Response: *we modified the sentence to acknowledge that this effect was observed in the mouse model we used and does not necessarily translate to human muscles (line 461, manuscript with tracked changes).*

4. When measuring muscle contractility in muscular dystrophy mouse studies, response to eccentric contraction is often included as a sensitive assay to detect the damage and rescue of function. Is there a reason for not doing this assay?

Response: *this analysis was indeed performed when we studied the effect of the combined AAV treatment in TA muscle strength. As reported in the figure below there no decrease in maximal force after 9 eccentric contractions is detected in A17 mice either treated with saline or with the combination of the vectors compared to muscles of FvB wild type muscles. This result was actually expected as the resistance to eccentric contraction is reduced only in dystrophic models where the expression of proteins of the dystrophin associated protein complex (DAPC) is somewhat compromised (e.g. in mdx mice where the lack of dystrophin depletes the DAPC of most of its proteins). TA muscles of A17 mice, although weaker, did not show reduction in maximal force generated after multiple lengthening contractions suggesting that the DAPC*

sustaining the sarcolemma is not affected by the pathology. Therefore we did not consider relevant to add these data to the manuscript.

5. Figure 1D western should include Flag blot to illustrate expPABPN1 expression.

Response: We agree with the reviewer that these data were missing. We have now added the immunoblot detecting the Flag epitope in Figure 1.

6. Figure 2H bottom panel images are barely visible.

Response: we increased the size of Figure 2H (now figure 2G).

7. The images used in the figure is of poor quality. Suggest to include large photos in Supplements.

Response: we increased the size of the pictures that are now more visible and we moved the graph showing the quantification of shRNA expressed in muscles as Supplementary figure 1B. Please note also that high resolution figures (which are too large for this submission) will be used for the final version once the manuscript is accepted for publication.

8. In Supp data, they provided shRNA3X construct sequence. It will be very helpful if the authors can mark the location of promoters and shRNA sequence use different color or different font etc.

Response: we modified the sequences in supplementary data to highlight the location of all the relevant regions of the constructs.

We thank the reviewer for his/her useful comments on the manuscript. The reviewer raised some criticisms that are addressed below.

The manuscript describes gene therapy in a mouse model of oculopharyngeal muscular dystrophy. The disease is a protein aggregation disease and surgery is currently the main treatment but is inadequate and patients die of complications of failure to swallow properly. Anti-aggregation small molecules and intrabodies are at various stages of testing but treatment targeting the genetic defect may provide permanent treatment. The authors have designed a two-vector in vivo strategy in the mice – to knock down the mutant expanded protein with one vector while introducing a knock-down resistant non-expanded gene. They do so using a tricistronic silencing vector alongside a gene supplementation vector. They show by various measures that local muscle pathology in the mouse model is improved by intramuscular injection of vector into the tibialis anterior. Commendably, they also demonstrate no toxic effect of injection of these vectors into wild type mice – toxicity that might arise, e.g., from full knock-down & insufficient correction, or from saturation of the molecular machinery involved in siRNA activity. Also, commendably, they test their system in human cells.

There are two specific, major concerns, which should be addressed.

1) No analysis of ectopic expression – both sides of the coin. Did the authors look for evidence of correction of pathology in distal muscles? More importantly, did the authors look for vector spread beyond the site of injection? Finally, and even more importantly, did the authors look at what happened when they intentionally delivered the vector non-locally i.e. by systemic injection?

2) How would one administer to the relevant muscles in humans? The authors allude to this, but do not detail how it might be achieved. Has this been performed with any substance, in humans or large animals, previously?

Response: *we made a common answer to the 2 points raised above by the reviewer. We did not analyse other muscles apart for the injected Tibialis anterior as in clinical setting only local intramuscular administration in affected muscles would be considered. All muscles of the A17 mouse model are affected by expPABPN1 expression and consequent intranuclear aggregates in myofibres. After intramuscular injection we expect that all (or almost all) the injected AAV vectors remain in the treated TA muscles. If some of the AAV leaks through the vasculature we expect that this would simply slightly improve the disease in the other transduced muscles reducing the amount of aggregates and maybe increasing muscle strength. We did not detect any toxic effect of the vector and mice were healthy at the end of the experiment. The reviewer is right that the outcome may change after a systemic injection of the AAV vectors as indeed we would expect toxic issues mainly associated to the expression of shRNA3X cassette into the liver. Of course, this is expected only if a significant amount of vector is delivered to the liver which is not the case with the doses and the route of delivery we used. However we do not consider the systemic delivery in mouse a relevant approach as, in human, the phenotype is only limited to specific muscles at least at the onset of disease (primarily pharyngeal muscles and muscles of eyes). Therefore the translation in clinical settings of the AAV mediated approach for OPMD would involve multiple local intramuscular injections (likely ranging from 1e09vp to 1e10vp per injection) performed only in affected muscles of patients that refer to doctors to undergo cricopharyngeal myotomy that is currently the only therapeutic (though*

not definitive) option. This approach based on local injections has been recently used for the clinical trial based on myoblast transplantation (NCT00773227) that showed a partial success (Perie et al, 2014).

3) Related to point (2) – are the authors sure that there are only very restricted muscles affected, or are they the only ones that kill the patient first? What happens when those are treated? Perhaps then other aggregations would occur in other muscles. Has anyone ever looked at post-mortem?

Response: *OPMD is clinically characterized by ptosis and dysphagia. Proximal limb weakness may occur at later stages of the disease. So only restricted muscles are clinically affected but aggregates and pathological features, such as rimmed vacuoles and ragged red fibers, are present in both clinically affected and clinically unaffected muscles from OPMD patients (Gidaro et al, 2013). So as regards to aggregation, this already occurs in all muscles.*

Minor comments:

Line 124/125 “the one” should probably read “that”

Response: *we modified the text accordingly (line 133, manuscript with tracked changes).*

Line 538 “intraperitoneally”...

Response: *we modified the text accordingly (line 581, manuscript with tracked changes).*

Figure 7 is really hard to understand – unfortunately a case of a picture not painting 1000 words. Use full words, describe more clearly on each part of the figure – maybe a brief narrative under each part/panel

Response: *We modified Figure 7 and we wrote a completely new legend to improve the description of all panels.*

References:

*Anvar SY, t’Hoen PA, Venema A, van der Sluijs B, van Engelen B, Snoeck M, Vissing J, Trollet C, Dickson G, Chartier A, Simonelig M, van Ommen GJ, van der Maarel SM, Raz V. Deregulation of the ubiquitin proteasome system is the predominant molecular pathology in OPMD animal models and patients. *Skeletal muscle*. 2011; 1:15.*

*Boldrin L, Zammit PS, Muntoni F, Morgan JE. Mature adult dystrophic mouse muscle environment does not impede efficient engrafted satellite cell regeneration and self-renewal. *Stem Cells*. 2009 Oct;27(10):2478-87.*

*Butler-Browne GS, Bugaisky LB, Cuénoud S, Schwartz K, Whalen RG. Denervation of newborn rat muscle does not block the appearance of adult fast myosin heavy chain. *Nature*. 1982; 28;299(5886):830-3.*

*Chartier A, Klein P, Pierson S, Barbezier N, Gidaro T, Casas T, Carberry S, Dowling P, Maynadier L, Bellec M, Oloko M, Jardel C, Moritz B, Dickson G, Mouly V, Ohlendieck K, Butler-Browne G, Trollet C, Simonelig M. Mitochondrial dysfunction reveals the role of mRNA poly(A) tail regulation in oculopharyngeal muscular dystrophy pathogenesis. *PLoS Genet*. 2015; 11, e1005092.*

Conacci-Sorrell M, Ngouenet C, Eisenman RN. Myc-nick: a cytoplasmic cleavage product of Myc that promotes alpha-tubulin acetylation and cell differentiation. *Cell*. 2010; 6;142(3):480-93.

De Greef JC, Hamlyn R, Jensen BS, O'CLanda, Levy JR, Kobuke K, Campbell KP. Collagen VI deficiency reduces muscle pathology, but does not improve muscle function, in the g-sarcoglycan-null mouse. *Hum Mol Gen*. 2016; 25(7): 1357-1369.

Dion P, Shanmugam V, Gaspar C, Messaed C, Meijer I, Toulouse A, Laganiere J, Roussel J, Rochefort D, Laganiere S, Allen C, Karpati G, Bouchard JP, Brais B, Rouleau GA. Transgenic expression of an expanded (GCG)₁₃ repeat PABPN1 leads to weakness and coordination defects in mice, *Neurobiol Dis*. 2005; 18(3): 528-536.

Gibertini S, Zanotti S, Savadori P, Curcio M, Saredi S, Salerno F, Andreetta F, Bernasconi P, Mantegazza R, Mora M. Fibrosis and inflammation are greater in muscles of beta-sarcoglycan-null mouse than mdx mouse. *Cell Tissue Res*. 2014;356(2):427-443

Gidaro T, Negroni E, Perie S, Mirabella M, Laine J, St Guily JL, Butler-Browne G, Mouly V, Trollet C. Atrophy, Fibrosis, and Increased PAX7-Positive Cells in Pharyngeal Muscles of Oculopharyngeal Muscular Dystrophy Patients. *J Neuropathol Exp Neurol*. 2013; 72, 234-243.

Malerba A, Sharp PS, Graham IR, Arechavala-Gomez V, Foster K, Muntoni F, Wells DJ, Dickson G. Chronic systemic therapy with low-dose morpholino oligomers ameliorates the pathology and normalizes locomotor behavior in mdx mice. *Mol Ther*. 2011; 19(2):345-54

Perie S, Trollet C, Mouly V, Vanneaux V, Mamchaoui K, Bouazza B, JP Marolleau, Laforêt P, Chapon F, Eymard B, Butler-Browne G, Larghero J, St Guily JL. *Mol Ther*. 2014; 22(1): 219-225

Veal EA, Jackson MJ. C-myc is expressed in mouse skeletal muscle nuclei during post-natal maturation. *Int J Biochem Cell Biol*. 1998; 30(7):811-21.

REVIEWERS' COMMENTS:

Reviewer #1 (Remarks to the Author):

The authors have made great efforts to answer all reservations of reviewers and modify extensively the manuscript to meet all comments, request for corrections and suggestions. I have no further reservations or comments to make. The paper is now well balanced and clearer.

Reviewer #2 (Remarks to the Author):

All my questions have been addressed. However, I'd strongly suggest to include the data from eccentric contraction in the supplementary materials. The differences in the pattern have important biological and physiological implications for the disease.

As the authors stated that other muscles in human patients do show pathology. It is very likely that after pharyngeal muscles are treated, one may start to see symptoms in other muscles due to increased survival (extended lifespan). In this case, a systemic therapy may still be needed.

Reviewer #3 (Remarks to the Author):

The authors have addressed all of my comments, and those of the other reviewers, comprehensively and thoughtfully.

In the spirit of transparency which this journal is supporting, I am happy to de-anonymise this review.

Simon Waddington

Author response to Reviewer #2 Comments: Manuscript NCOMMS-16-16130A, Malerba et al, PABPN1 gene therapy rescues disease phenotype in oculopharyngeal muscular dystrophy

All my questions have been addressed. However, I'd strongly suggest to include the data from eccentric contraction in the supplementary materials. The differences in the pattern have important biological and physiological implications for the disease. **Response:** *The data showing that resistance to lengthening eccentric contractions is not affected in the A17 mice as compared the FvB control strain has been added as Supplementary Figure 4.*

As the authors stated that other muscles in human patients do show pathology. It is very likely that after pharyngeal muscles are treated, one may start to see symptoms in other muscles due to increased survival (extended lifespan). In this case, a systemic therapy may still be needed.

Response: *Generally, OPMD is initially characterized by ptosis and dysphagia, and it is these symptoms that most distress patients. However variable involvement of proximal limb and other musculature does occur, especially as the disease progresses. While treatment of swallowing muscles is a clear primary target, we agree with the reviewer that systemic gene therapy might be a future consideration. Clinically it should be noted that the systemic approach raises important but not insurmountable issues of vector manufacture and heightened safety concerns and then the systemic approach would be initially tested in the OPMD mouse model.*